# AAV-Txnip prolongs cone survival and vision in mouse models of retinitis pigmentosa

Yunlu Xue[1,2], Sean K Wang[1,2,3], Parimal Rana[1], Emma R West[1,3], Christin M Hong[1,3], Helian Feng[4], David M Wu[1,2,5], Constance L Cepko[1,2,3]*

[1]Department of Genetics, Blavatnik Institute, Harvard Medical School, Boston, United States; [2]Department of Ophthalmology, Harvard Medical School, Boston, United States; [3]Howard Hughes Medical Institute, Chevy Chase, United States; [4]Department of Biostatistics, Harvard T.H. Chan School of Public Health, Boston, United States; [5]Retina Service, Massachusetts Eye and Ear Infirmary, Harvard Medical School, Boston, United States

**Abstract** Retinitis pigmentosa (RP) is an inherited retinal disease affecting >20 million people worldwide. Loss of daylight vision typically occurs due to the dysfunction/loss of cone photoreceptors, the cell type that initiates our color and high-acuity vision. Currently, there is no effective treatment for RP, other than gene therapy for a limited number of specific disease genes. To develop a disease gene-agnostic therapy, we screened 20 genes for their ability to prolong cone photoreceptor survival in vivo. Here, we report an adeno-associated virus vector expressing Txnip, which prolongs the survival of cone photoreceptors and improves visual acuity in RP mouse models. A *Txnip* allele, C247S, which blocks the association of Txnip with thioredoxin, provides an even greater benefit. Additionally, the rescue effect of Txnip depends on lactate dehydrogenase b (Ldhb) and correlates with the presence of healthier mitochondria, suggesting that Txnip saves RP cones by enhancing their lactate catabolism.

*For correspondence:
cepko@genetics.med.harvard.edu

**Competing interests:** The authors declare that no competing interests exist.

## Introduction

Retinitis pigmentosa (RP) is one of the most prevalent types of inherited retinal diseases affecting approximately 1 in ~4,000 people (*Hartong et al., 2006*). In RP, the rod photoreceptors, which initiate night vision, are primarily affected by the disease genes and degenerate first. The degeneration of cones, the photoreceptors that initiate daylight, color, and high-acuity vision, then follows, which greatly impacts the quality of life. Currently, one therapy that holds great promise for RP is gene therapy using adeno-associated virus (AAV) (*Maguire et al., 2019*). This approach has proven successful for a small number of genes affecting a few disease families (*Cehajic-Kapetanovic et al., 2020*). However, due to the number and functional heterogeneity of RP disease genes (≈ 100 genes that primarily affect rods, https://sph.uth.edu/retnet/), gene therapy for each RP gene will be logistically and financially difficult. In addition, a considerable number of RP patients do not have an identified disease gene. A disease gene-agnostic treatment aimed at prolonging cone function/survival in the majority of RP patients could thus benefit many more patients. Given that the disease gene is typically not expressed in cones, and thus their death is due to non-autonomous mechanisms that may be in common across affected families, answers to the question of why cones die may provide an avenue to a widely applicable therapy for RP. To date, the suggested mechanisms of cone death include oxidative damage (*Komeima et al., 2006*; *Wellard et al., 2005*; *Xiong et al., 2015*), inflammation (*Wang et al., 2020*; *Wang et al., 2019*; *Zhao et al., 2015*), and a shortage of nutrients (*Aït-Ali et al., 2015*; *Kanow et al., 2017*; *Punzo et al., 2012*; *Punzo et al., 2009*; *Wang et al., 2016*).

**eLife digest** Retinitis pigmentosa is an inherited eye disease affecting around one in every 4,000 people. It results from genetic defects in light sensitive cells of the retina, called photoreceptor cells, which line the back of the eye. Though vision loss can occur from birth, retinitis pigmentosa usually involves a gradual loss of vision, sometimes leading to blindness. Rod photoreceptors, which are responsible for vision in low light, are impacted first. The disease then affects cone photoreceptors, the cells that detect light during the day, providing both color and sharp vision.

Around 100 mutated genes associated with retinitis pigmentosa have been identified, but only a handful of families with one of these mutant genes have been treated with a gene therapy specific for their mutated gene. There are currently no therapies available to treat the vast number of people with this disease. The mutations that cause retinitis pigmentosa directly affect the rod cells that detect dim light, leading to loss of night vision. There is also an indirect effect that causes cone photoreceptors to stop working and die. One theory to explain this two-step disease process relates to the fact that cone photoreceptors are very active cells, requiring a high level of energy, nutrients and oxygen. If surrounding rod cells die, cone photoreceptors may be deprived of some essential supplies, leading to cone cell death and daylight vision loss.

To examine this theory, Xue et al. tested a new gene therapy designed to alleviate the potential shortfall in nutrients. The experiments used three different strains of mice that had the same genetic mutations as humans with retinitis pigmentosa. The gene therapy used a virus, called adeno-associated virus (AAV), to deliver 20 different genes to cone cells. Each of the 20 genes tested plays a different role in cells' processing of nutrients to provide energy. After administering the treatment, Xue et al. monitored the mice to see whether or not their vision was affected, and how cone cells responded.

Only one of the 20 genes, *Txnip*, delivered using gene therapy, had a beneficial effect, prolonging cone cell survival in all three mouse strains. The mice that received *Txnip* also retained their ability to discern moving stripes on vision tests. Further investigations demonstrated that activating *Txnip* forced the cones to start using a molecule called lactate as an energy source, which could be more available to them than glucose, their usual fuel. These cells also had healthier mitochondria – the compartments inside cells that produce and manage energy supplies. This dual effect on fuel use and mitochondrial health is thought to be the basis for the extended cone survival and function.

These experiments by Xue et al. have identified a good gene therapy candidate for treating retinitis pigmentosa independently of which genes are causing the disease. Further research will be required to test the safety of the gene therapy, and whether its beneficial effects translate to humans with retinitis pigmentosa, and potentially other diseases with unhealthy photoreceptors.

In 2009, we surveyed gene expression changes that occurred during retinal degeneration in four mouse models of RP (*Punzo et al., 2009*). Those data led us to suggest a model wherein cones starve and die due to a shortage of glucose, which is typically used for energy and anabolic needs in photoreceptors via glycolysis. Evidence of this 'glucose shortage hypothesis' was subsequently provided by orthogonal approaches from other groups (*Aït-Ali et al., 2015*; *Wang et al., 2016*). These studies have inspired us to test 20 genes that might affect the uptake and/or utilization of glucose by cones in vivo in three mouse models of RP (*Figure 1—source data 1*). Only one gene, *Txnip*, had a beneficial effect, prolonging cone survival and visual acuity in these models. *Txnip* encodes an α-arrestin family member protein with multiple functions, including binding to thioredoxin (*Junn et al., 2000*; *Nishiyama et al., 1999*), facilitating removal of the glucose transporter 1 (GLUT1) from the cell membrane (*Wu et al., 2013*), and promoting the use of non-glucose fuels (*DeBalsi et al., 2014*). Because α-arrestins are structurally distinct from the visual or β-arrestins, such as ARR3, Txnip is unlikely to bind to opsins or to participate in phototransduction (*Hwang et al., 2014*; *Kang et al., 2015*; *Puca and Brou, 2014*). We tested a number of *Txnip* alleles and found that one allele, C247S, which blocks the association of Txnip with thioredoxin (*Patwari et al., 2006*), provided the greatest benefit. Investigation of the mechanism of Txnip rescue revealed that it required lactate dehydrogenase b (Ldhb), which catalyzes the conversion of lactate to pyruvate. Imaging of metabolic reporters

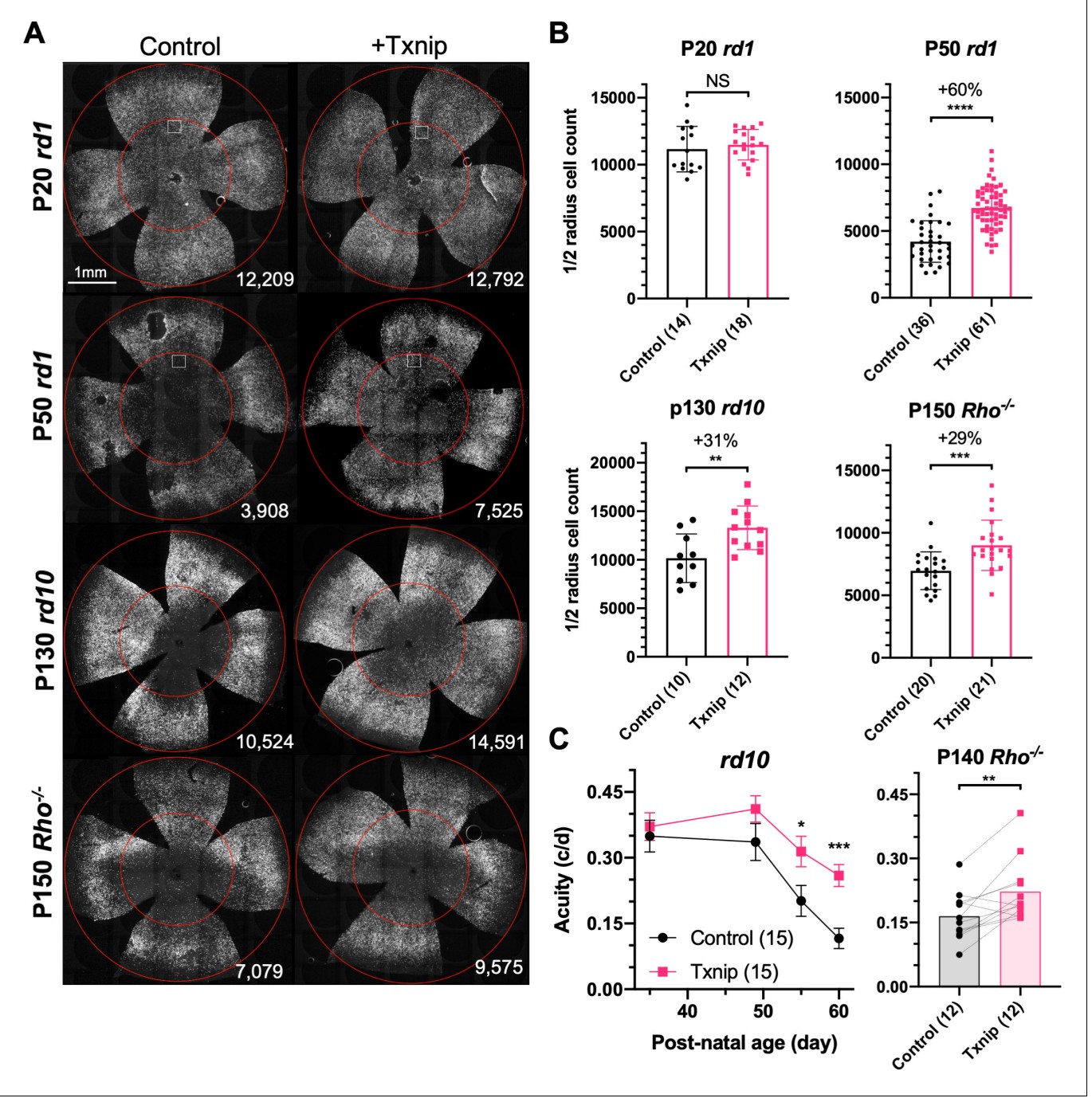

**Figure 1.** Txnip effects on cone survival and cone-mediated vision in retinitis pigmentosa (RP) mice. (**A**) Representative images from postnatal day 20 (P20) and P50 *rd1*, P130 *rd10*, and P150 *Rho*⁻/⁻ flat-mounted retinas, in which retinas were infected with adeno-associated viruses (AAVs) encoding Txnip and H2BGFP (AAV8-RedO-Txnip, ≈1 × 10⁹ vg/eye plus AAV8-RedO-H2BGFP, 2.5 × 10⁸ vg/eye) or control (AAV8-RedO-H2BGFP, 2.5 × 10⁸ vg/ eye). The outer circle was drawn to mark the outline of the retina, and the inner circle was drawn to the ½ radius of the outer circle. The small boxes in the top four panels mark the regions shown at higher magnification in *Figure 1—figure supplement 1C*, demonstrating the pixels recognized as cones by the MATLAB automated-counting program. The number at the lower-right corner in each panel is the count of cones within the ½ radius of each image. All H2BGFP-labeled cones were counted within the central retina defined by the ½ radius (i.e., not just the cells from the small boxes). (**B**) Quantification of H2BGFP-positive cones within the ½ radius of the retina for different groups (same as in **A**). Error bar: standard deviation. The number in the round brackets '()' indicates the sample size, that is, the number of retinas within each group. (**C**) Visual acuity of *rd10* and *Rho*⁻/⁻ mice transduced with Txnip or H2BGFP alone in each eye measured using an optomotor assay. Error bar: SEM. NS: not significant; p>0.05, *p<0.05, **p<0.01, ***p<0.001, **** p< or <<0.0001. RedO: red opsin promoter; AAV: adeno-associated virus.

*Figure 1 continued on next page*

*Figure 1 continued*

The online version of this article includes the following source data and figure supplement(s) for figure 1:

**Source data 1.** Adeno-associated virus 8 vectors used in this study.
**Figure supplement 1.** Additional figures for effects of metabolic genes on cone survival.
**Figure supplement 2.** Additional figures for Txnip effects on cone survival.

demonstrated an enhanced intracellular ATP:ADP ratio when the retina was placed in lactate medium. Moreover, by several measures, mitochondria appeared to be healthier as a result of Txnip addition, but this improvement was not sufficient for cone rescue.

The above observations led to a model wherein Txnip shifts cones from their normal reliance on glucose to enhanced utilization of lactate, as well as marked improvement in mitochondrial structure and function. Analysis of the rescue activity of several additional genes predicted to affect glycolysis provided support for this model. Finally, as our goal is to rescue cones that suffer not only from metabolic challenges, but also from inflammation and oxidative damage, we tested Txnip in combination with anti-inflammatory and anti-oxidative damage genes, and found additive benefits for cones. These treatments may benefit cones not only in RP, but also in other ocular diseases where similar environmental stresses are present, such as in age-related macular degeneration (AMD).

## Results

### Txnip prolongs RP cone survival and visual acuity

We delivered genes that might address a glucose shortage and/or mismanagement of metabolism in a potentially glucose-limited environment. To this end, 12 AAV vectors were constructed to test genes singly or in combination for an initial screen (*Figure 1—figure supplement 1E*). Subsequently, an additional set of AAV vectors were made based upon the initial screen results, as well as other rationales, to total 20 genes tested in all (*Figure 1—source data 1*). Most of these vectors carried genes to augment the utilization of glucose, such as hexokinases, phosphofructokinase, and pyruvate kinase. Each AAV vector used a cone-specific promoter, which was previously found to be non-toxic at the doses used in this study (*Xiong et al., 2019*). An initial screen was carried out in *rd1* mice, which harbor a null allele in the rod-specific gene, *Pde6b*. This strain has a rapid loss of rods, followed by cone death. The vectors were subretinally injected into the eyes of neonatal *rd1* mice, in combination with a vector using the human red opsin (RedO) promoter, to express a histone 2B-GFP fusion protein (AAV-RedO-H2BGFP). The H2BGFP provides a very bright cone-specific nuclear labeling, enabling automated quantification. As a control, eyes were injected with AAV-RedO-H2BGFP alone. *Rd1* cones begin to die at ≈postnatal day 20 (P20) after almost all rods have died (*Figure 1—figure supplement 1A*, *Figure 1—figure supplement 2A*). The number of *rd1* cones was quantified by counting the H2BGFP+ cells using a custom-made MATLAB program (*Figure 1A*, *Figure 1—figure supplement 1C*). Because ~11,000 *rd1* cones were counted in the central ½ radius of retina before their death at P20 (*Figure 1—figure supplement 1E*), we estimated ~20% H2BGFP labeling efficiency using data for wildtype mice for comparison (i.e., ~50,000 cones within ½ radius of wild-type retina) (*Jeon et al., 1998*), with this injection dose. Only cones within the central ½ radius region of the retina were counted since RP cones in the periphery die much later (*Hartong et al., 2006*; *Punzo et al., 2009*). Among the vectors with individual or combinations of genes, only Txnip preserved *rd1* cones at P50 (*Figure 1A, B*, *Figure 1—figure supplement 1E*). The effects were likely on cone survival as it did not change the number of cones at P20 prior to their death, but did provide survival benefit by ~P30 (*Figure 1A, B*, *Figure 1—figure supplement 2A–C*). The level of Txnip rescue in P50 *rd1* cones was comparable to that seen using AAV with a cytomegalovirus (CMV) promoter to express a transcription factor, Nrf2, that regulates anti-oxidation pathways and reduces inflammation, as we found previously (*Xiong et al., 2015*; *Figure 1—figure supplement 1E*). One combination led to a reduction in cone survival, that of Hk1 plus Pfkm (*Figure 1—figure supplement 1E*).

Our initial screen used the RedO promoter to drive Txnip expression. To evaluate a different cone-specific promoter, Txnip also was tested using a newly described cone-specific promoter, Syn-PVI (*Jüttner et al., 2019*). This promoter also led to prolonged cone survival (*Figure 1—figure*

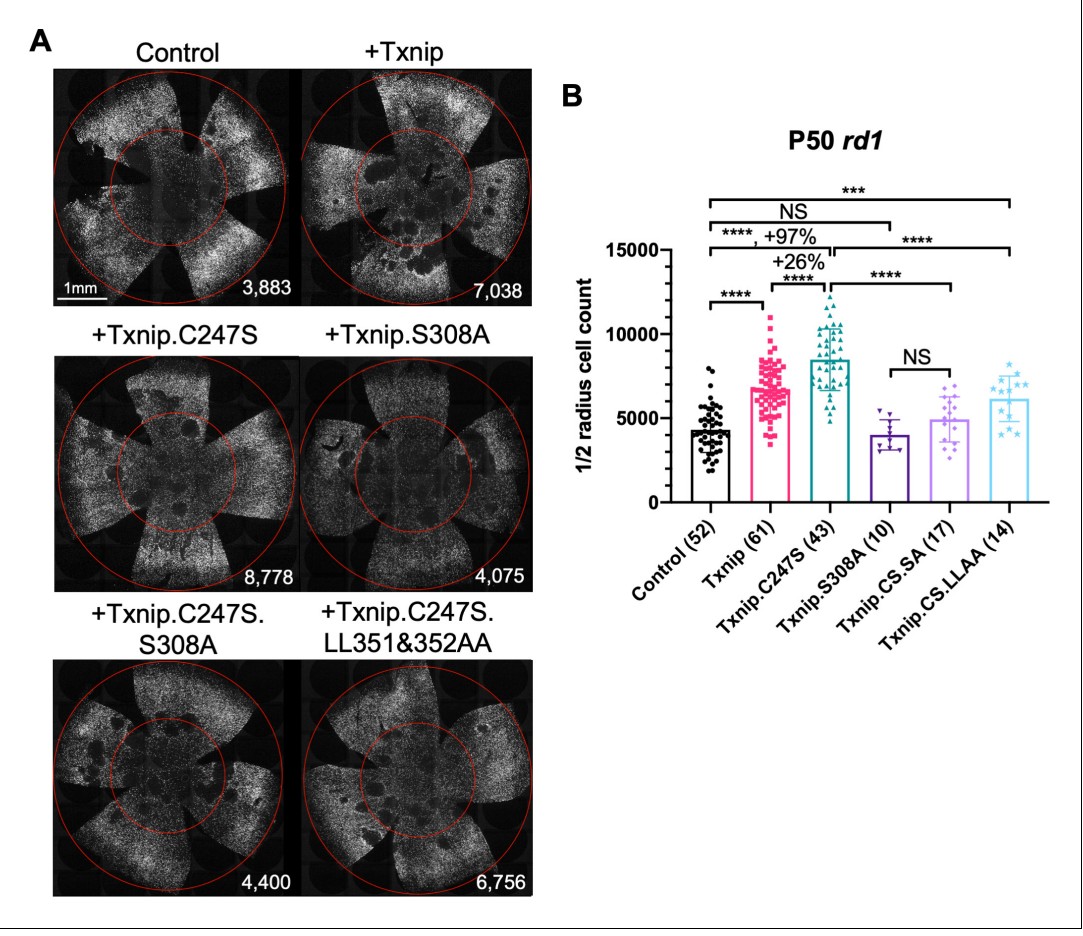

**Figure 2.** Test of Txnip alleles on cone survival. (**A**) Representative P50 *rd1* flat-mounted retinas after P0 infection with one of five different Txnip alleles (AAV8-RedO-Txnip wildtype (WT)/.C247S/.S308A/.C247S.S308A/.C247S.LL351 and 352AA, ≈1 × 10$^9$ vg/eye, plus AAV8-RedO-H2BGFP, 2.5 × 10$^8$ vg/ eye), or control eyes infected with AAV8-RedO-H2BGFP, 2.5 × 10$^8$ vg/eye alone. (**B**) Quantification of H2BGFP-positive cones within the ½ radius of P50 *rd1* retinas transduced with WT Txnip, Txnip alleles, and control (same as in **A**). The number in the round brackets '()' indicates the sample size, that is, the number of retinas within each group. Error bar: standard deviation. Txnip.CS.SA: Txnip.C247S.S308A; Txnip.CS.LLAA: Txnip.C247S.LL351 and 352AA. NS: not significant, p>0.05, *p<0.05, **p<0.01, ***p<0.001, **** p< or <<0.0001. RedO: red opsin promoter; AAV: adeno-associated virus. The online version of this article includes the following figure supplement(s) for figure 2:

**Figure supplement 1.** Additional figures for effects of Txnip alleles on cone survival.

**Figure supplement 2.** Slc2a1/GLUT1 shRNA in vitro screening.

*supplement 1E*). To explore whether Txnip gene therapy is effective beyond *rd1*, it was tested in *rd10* mice, which carry a missense *Pde6b* mutation, and in *Rho*$^{-/-}$ mice, which carry a null allele in a rod-specific gene, rhodopsin. Cone survival was evaluated after the majority of central cones had died, with different ages for different strains, based upon our previous work (*Punzo et al., 2009*; *Wang et al., 2019*; *Xiong et al., 2015*). Both *rd10* and *Rho*$^{-/-}$ mice showed improved cone survival (*Figure 1A, B*). The rescue effect did not persist long term, however, as by P240 in the *Rho*$^{-/-}$ strain it was not significant (*Figure 1—figure supplement 2D*). To determine if Txnip-transduced mice sustained greater visual performance than control RP mice, an optomotor assay was used to measure maximal visual threshold for spatial frequency (i.e., visual acuity) (*Prusky et al., 2004*). Under conditions that simulated daylight, Txnip-transduced eyes showed enhanced visual acuity compared to the control contralateral eyes in *rd10* and *Rho*$^{-/-}$ mice (*Figure 1C*). The *rd1* strain degenerates so quickly that it could not be evaluated in this assay. To determine if there was an improvement in overall cone phototransduction, summed across all cones, electroretinography (ERG) was carried out. No effect was observed in *rd10* mice transduced with Txnip (*Figure 1—figure supplement 2E*). Txnip also was evaluated for effects on cones in wildtype (WT) mice using peanut agglutinin (PNA)

staining, which stains the cone-specific extracellular matrix and reflects cone health. No effect was seen on PNA staining (*Figure 1—figure supplement 1D*). In addition, retinas from both WT and P21 *rd1* mice were stained using anti-ARR3, which stains the entire cone. At P31, the approximate number and morphology of Txnip-transduced cones in WT retinas was similar to uninfected WT retinas (*Figure 1—figure supplement 2A*). At P21 and P30, immunohistochemistry (IHC) for ARR3 in *rd1* retinas did not show an obvious rescue of cone outer segments by Txnip (*Figure 1—figure supplement 2A*).

## Evaluation of *Txnip* alleles for cone survival

Previous studies of Txnip provided a number of alleles that could potentially lead to a more effective cone rescue by Txnip and/or provide some insight into which of the Txnip functions are required for enhancing cone survival. A C247S mutation has been shown to block Txnip's inhibitory interaction with thioredoxin (*Patwari et al., 2009*; *Patwari et al., 2006*), which is an important component of a cell's ability to fight oxidative damage via thiol groups (*Junn et al., 2000*; *Nishinaka et al., 2001*; *Nishiyama et al., 1999*). If cone rescue by Txnip required this function, the C247S allele should be less potent for cone rescue. Alternatively, if loss of thioredoxin binding freed Txnip for its other functions and made more thioredoxin available for oxidative damage control, this allele might more effectively promote cone survival. The C247S clearly provided more robust cone rescue than WT Txnip in all three RP mouse strains (*Figure 2*, *Figure 2—figure supplement 1A, B*). These results indicate that the therapeutic effect of Txnip does not require an inhibitory interaction with thioredoxins. This finding is in keeping with previous work, which showed that anti-oxidation strategies promoted cone survival in RP mice (*Komeima et al., 2006*; *Wu et al., 2021*; *Xiong et al., 2015*). An additional mutation, S308A, which loses an AMPK/Akt-phosphorylation site on Txnip (*Waldhart et al., 2017*; *Wu et al., 2013*), was tested in the context of WT Txnip and in the context of the C247S allele. The S308A change did not benefit cone survival in either context (*Figure 2*). In addition, the S308A allele was assayed for negative effects on cones by an assessment of *rd1* cone number prior to P20, that is, before the onset of cone death (*Figure 2—figure supplement 1C*). It did not reduce the cone number at this early timepoint, indicating that Txnip.S308A was not toxic to cones. This finding suggests that the S308 residue is critical for the therapeutic function of Txnip through an unclear mechanism. One additional set of amino acid changes, LL351 and 352AA, was tested in the context of C247S. This allele eliminates a clathrin-binding site, and thus hampers Txnip's ability to remove GLUT1 from cell surface through clathrin-coated pits (*Wu et al., 2013*). Txnip.C247S.LL351 and 352AA could still delay RP cone death compared to the control (*Figure 2B*), suggesting that the therapeutic effect of Txnip was unlikely to be only through the removal of GLUT1 from the cell surface. To further explore the role of GLUT1, an shRNA to *Slc2a1*, which encodes GLUT1, was tested. It did not prolong RP cone survival (*Figure 2—figure supplement 1D*). The slight decrease of Txnip.C247S.LL351 and 352AA in cone rescue compared to Txnip.C247S might be due to other, currently unknown effects of LL351 and 352, or a less specific effect, for example, a protein conformational change.

## Txnip requires Ldhb to prolong cone survival

People with Txnip null mutations present with lactic acidosis (*Katsu-Jiménez et al., 2019*), suggesting that Txnip deficiency might compromise lactate catabolism. A metabolomic study of muscle using a targeted knockout of Txnip suggested that Txnip increases the catabolism of non-glucose fuels, such as lactate, ketone bodies, and lipids (*DeBalsi et al., 2014*). This switch in fuel preference was proposed to benefit the mitochondrial tricarboxylic acid cycle (TCA cycle), leading to a greater production of ATP. As presented earlier, a problem for cones in the RP environment might be a shortage of glucose (*Aït-Ali et al., 2015*; *Punzo et al., 2009*; *Wang et al., 2016*). A benefit of Txnip might then be to enable and/or force cells to switch from a preference for glucose to one or more alternative fuels. To test this hypothesis, we co-injected AAV-Txnip with shRNAs targeting the mRNAs for the rate-limiting enzymes for the catalysis of lactate, ketones, or lipids. Ldhb, encoded by the *Ldhb* gene, is the enzyme that converts lactate to pyruvate to potentially fuel the TCA cycle, and lactate dehydrogenase a (Ldha, encoded by *Ldha* gene) converts pyruvate to lactate (*Eventoff et al., 1977*). We found that Txnip rescue was significantly decreased by any one of three Ldhb shRNAs (siLdhb) or by overexpression of Ldha (*Figure 3A, B*, *Figure 3—figure supplement*

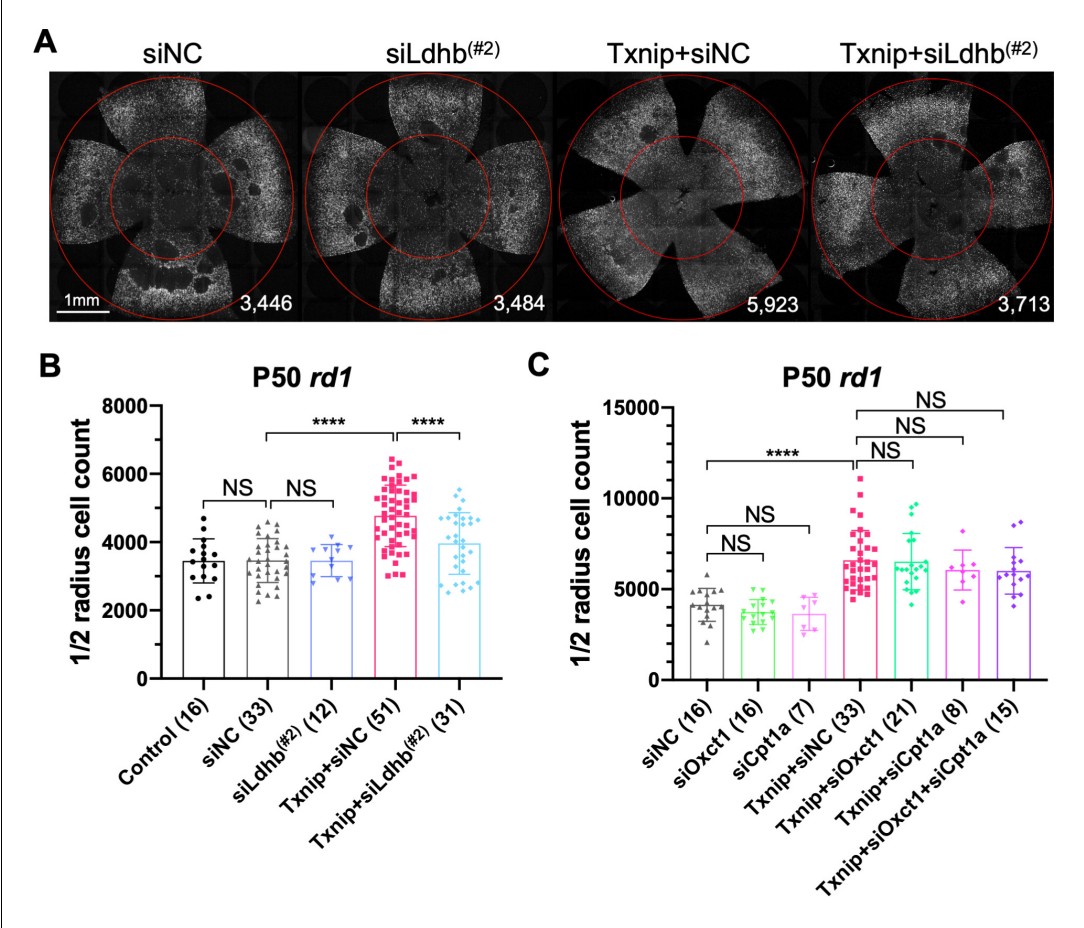

**Figure 3.** Effect of knockdown of lactate dehydrogenase b (Ldhb) in Txnip-transduced retinitis pigmentosa cones in vivo. (A) Representative P50 *rd1* flat-mounted retinas after P0 infection with control shRNA construct (siNC) or an shRNA construct targeting Ldhb (siLdhb[(#2)]) in the presence or absence of transduced Txnip (AAV8-RedO-Txnip, ≈1 × 10⁹ vg/eye; AAV8-RedO-shRNA ≈1 × 10⁹ vg/eye), plus AAV8-RedO-H2BGFP (2.5 × 10⁸ vg/eye) (B) Quantification of H2BGFP-positive cones within the ½ radius of P50 *rd1* retinas transduced with control, siNC control, Txnip + siLdhb[(#2)], or Txnip + siNC control (same as in A). (C) Quantification of H2BGFP-positive cones within the ½ radius of P50 *rd1* retinas transduced with Txnip + siOxct1[(#c)], Txnip + siCpt1a[(#c)], Txnip + siOxct1[(#c)] + siCpt1a[(#c)], or siNC control. (All are AAV8-RedO-Txnip, ≈1 × 10⁹ vg/eye; AAV8-RedO-shRNA, ≈1 × 10⁹ vg/eye; plus AAV8-RedO-H2BGFP, 2.5 × 10⁸ vg/eye.) Error bar: standard deviation. NS: not significant, p>0.05, **p<0.01, ***p<0.001, **** p< or <<0.0001. RedO: red opsin promoter; AAV: adeno-associated virus.

The online version of this article includes the following figure supplement(s) for figure 3:

**Figure supplement 1.** Additional figures for the dependency of Txnip rescue on lactate dehydrogenase b and effect of Ldha.

**Figure supplement 2.** Lactate dehydrogenase b (Ldhb) shRNA in vitro screening.

**Figure supplement 3.** Oxct1 shRNA in vitro screening.

**Figure supplement 4.** Cpt1a shRNA in vitro screening.

1B–E). We also tested the rescue effect of Txnip plus an shRNA against Oxct1 (siOxct1), a critical enzyme for ketolysis (*Zhang and Xie, 2017*), or against Cpt1a (siCpt1a), a component for lipid transporter that is rate limiting for β-oxidation (*Shriver and Manchester, 2011*). These shRNAs, tested singly or in combination, did not reduce the effectiveness of Txnip rescue (*Figure 3C*). Taken together, these data support the use of lactate, but not ketones or lipids, as a critical alternative fuel for cones when Txnip is overexpressed.

## Txnip improves the ATP:ADP ratio in RP cones in the presence of lactate

If the improved survival of cones following Txnip overexpression is due to improved utilization of non-glucose fuels, cones might show improved mitochondrial metabolism. To begin to examine the

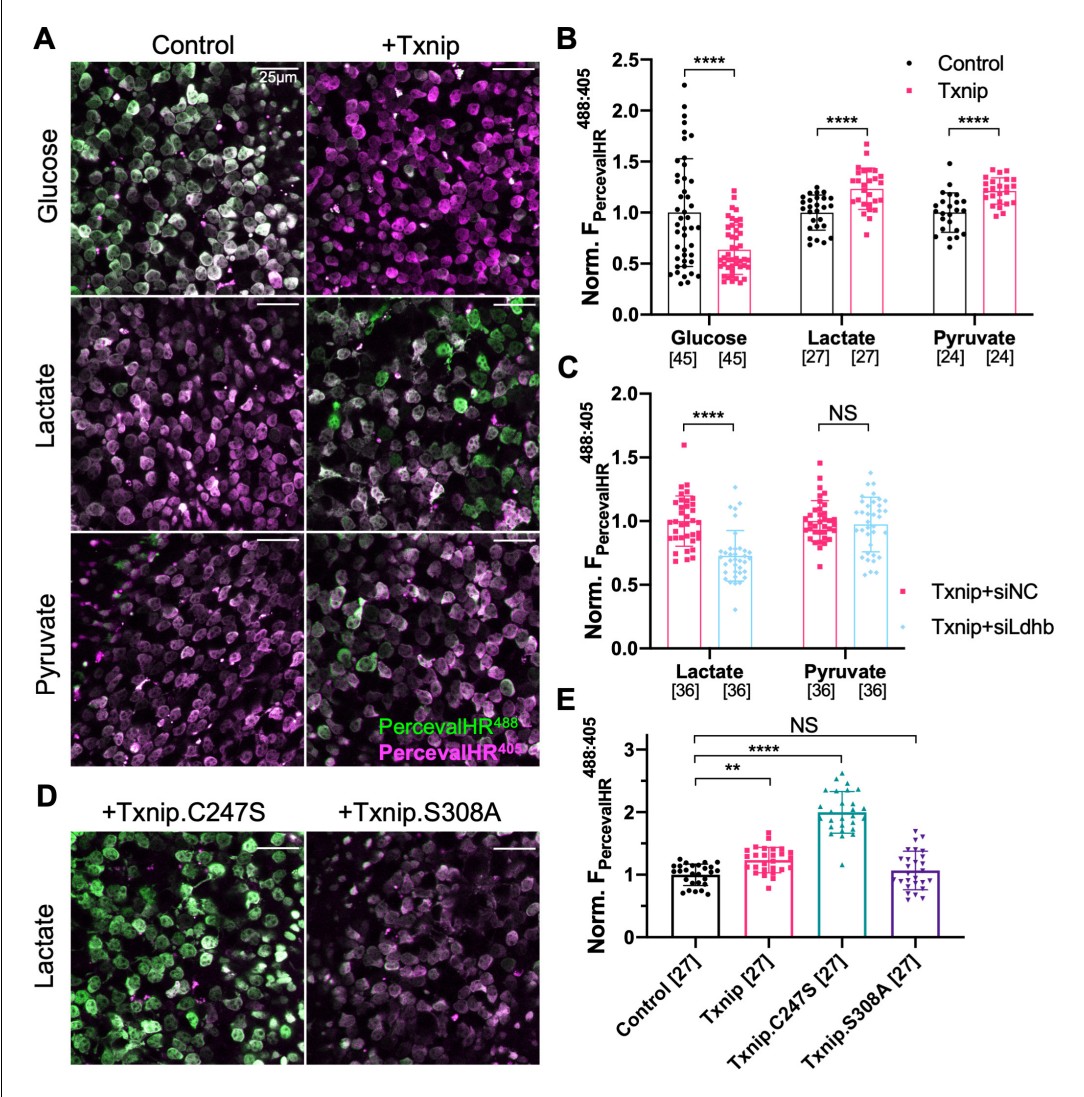

**Figure 4.** Effect of Txnip on ATP:ADP levels in retinitis pigmentosa (RP) cones in media with different carbon sources. (**A**) Representative ex vivo live images of PercevalHR-labeled cones in P20 *rd1* retinas cultured with high-glucose, lactate-only, or pyruvate-only medium and transduced with Txnip (AAV8-RedO-Txnip, $1 \times 10^9$ vg/eye, plus AAV8-RO1.7-PercevalHR, $1 \times 10^9$ vg/eye) (RO1.7 is a shorter version of the red opsin [RedO] promoter with a similar expression pattern) or control (i.e., AAV8-RO1.7-PercevalHR, $1 \times 10^9$ vg/eye). Magenta: fluorescence by 405 nm excitation, indicating low-ATP: ADP; green: fluorescence by 488 nm excitation, indicating high-ATP:ADP. (**B**) Quantification of normalized PercevalHR fluorescence intensity ratio ($F_{PercevalHR}^{ex488nm:\ ex405nm}$, proportional to ATP:ADP ratio) in cones from P20 *rd1* retinas in different conditions. The number in the square brackets '[]' indicates the sample size, that is, the number of images taken from regions of interest of multiple retinas (≈3 images per retina), in each condition. (**C**) Quantification of normalized PercevalHR fluorescence intensity of retinas infected with Txnip + siLdhb[#2] and Txnip + siNC in cones from P20 *rd1* retina in lactate-only or pyruvate-only medium. (AAV8-RedO-Txnip, ≈1 × $10^9$ vg/eye; AAV8-RedO-shRNA ≈1 × $10^9$ vg/eye; plus AAV8-RO1.7-PercevalHR, 1 × $10^9$ vg/eye.) (**D**) Representative ex vivo live images of PercevalHR-labeled cones in P20 *rd1* retinas cultured in lactate-only medium, following transduction with Txnip.C247S (AAV8-RedO-Txnip.C247S, 1 × $10^9$ vg/eye) or Txnip.S308A (AAV8-RedO-Txnip.S308A, 1 × $10^9$ vg/eye). Magenta: fluorescence by 405 nm excitation, indicating low-ATP:ADP; green: fluorescence by 488 nm excitation, indicating high-ATP:ADP. (**E**) Quantification of normalized PercevalHR fluorescence intensity following transduction by Txnip, Txnip alleles, and control cones in *P20 rd1* retinas cultured in lactate-only medium. Error bar: standard deviation. NS: not significant, p>0.05, **p<0.01, ***p<0.001, **** p< or <<0.0001. AAV: adeno-associated virus.

The online version of this article includes the following figure supplement(s) for figure 4:

**Figure supplement 1.** Effect of Txnip on pH and glucose levels in retinitis pigmentosa (RP) cones.

metabolism of cones, we first attempted to perform metabolomics of cones with and without Txnip. However, so few cones are present in these retinas that we were unable to achieve reproducible results. An alternative assay was conducted to measure the ratio of ATP to ADP using a

genetically encoded fluorescent sensor (GEFS). AAV was used to deliver PercevalHR, an ATP:ADP GEFS (*Tantama et al., 2013*), to *rd1* cones with and without AAV-Txnip. The infected P20 *rd1* retinas were explanted and imaged in three different types of media to measure the cone intracellular ratio of ATP:ADP. Txnip increased the ATP:ADP ratio (i.e., higher $F_{PercevalHR}^{488:405}$) of *rd1* cones in lactate-only medium. This was also seen in pyruvate-only medium, perhaps due to improved mitochondrial health (i.e., greater oxidative phosphorylation [OXPHOS] activity). Consistent with the role of Txnip in removing GLUT1 from the plasma membrane, Txnip-transduced cones had a lower ATP:ADP ratio (i.e., lower $F_{PercevalHR}^{488:405}$) in high-glucose medium (*Figure 4A, B*). To further probe whether intracellular glucose was reduced after overexpression of Txnip (*Wu et al., 2013*), a glucose sensor iGlucoSnFR was used (*Keller et al., 2019*). This sensor showed reduced intracellular glucose in Txnip-transduced cones (*Figure 4—figure supplement 1A, B*). Because the fluorescence of GEFS may also be subject to environmental pH, we used a pH sensor, pHRed (*Tantama et al., 2011*), to determine if the changes of PercevalHR and iGlucoseSnFR were due to a change in pH, and found no significant pH change (*Figure 4—figure supplement 1C, D*). We also found that lactate, but not pyruvate, utilization by Txnip-transduced cones was critically dependent upon Ldhb for ATP production as introduction of siLdhb abrogated the increase in ATP:ADP in Txnip-transduced cones (*Figure 4C*). Furthermore, in correlation with improved cone survival by Txnip.C247S compared to WT Txnip (*Figure 2B*), cones had a higher ATP:ADP ratio in lactate medium when Txnip.C247S was used relative to WT Txnip (*Figure 4D, E*). Similarly, in correlation with no survival benefit when transduced with Txnip.S308A (*Figure 2B*), there was no difference in the ATP:ADP ratio when Txnip.S308A was used, relative to control, in lactate medium (*Figure 4D, E*).

## Txnip improves RP cone mitochondrial gene expression, size, and function

To further probe the mechanism(s) of Txnip rescue, we first tested if all of the benefits of Txnip were due to Txnip's effects on Ldhb. Ldhb was thus overexpressed alone or with Txnip. Ldhb alone did not prolong cone survival, nor did it increase the Txnip rescue (*Figure 8—figure supplement 1D*). An additional experiment was carried out to investigate if there might be a shortage of the mitochondrial pyruvate carrier, which could limit the uptake of pyruvate into the mitochondria of photoreceptors for ATP synthesis (*Grenell et al., 2019*). The pyruvate carrier, which is a dimer encoded by *Mpc1* and *Mpc2* genes, was overexpressed, but did not prolong *rd1* cone survival (*Figure 7—figure supplement 1C*). To take a less biased approach, the transcriptomic differences between Txnip-transduced and control RP cones were characterized. H2BGFP-labeled RP cones were isolated by FACS sorting at an age when cones were beginning to die, and RNA sequencing was performed (*Figure 5—figure supplement 1A*). Data were obtained from two RP strains, *rd1* and *Rho*^-/-. By comparing the differentially expressed genes in common between the two strains, relative to control, 7 genes were seen to be upregulated and 17 were downregulated (*Figure 5—source data 1*). Three of the seven upregulated genes were mitochondrial electron transport chain (ETC) genes. The upregulation of these three ETC genes in Txnip-transduced *rd1* cones was confirmed by ddPCR (*Figure 5—figure supplement 1B*). Similarly, we also looked for transcriptomic differences induced by Txnip in WT cones using C57BL/6J and BALB/c mice, and found only Txnip mRNA upregulation in common (*Figure 5—figure supplement 2A*, *Figure 5—source data 2*). Interestingly, there was almost no Txnip mRNA detected by RNA-seq in the WT or RP control cones, but there was high number of Txnip transcripts following addition of RedO-Txnip in all strains (*Figure 5—source data 3*).

The finding of upregulated ETC genes in Txnip-transduced RP cones, but not in WT cones, suggested effects of Txnip on mitochondria during cone degeneration. Murakami et al. previously showed that cone mitochondria were swollen and deteriorated in *rd10* mouse retinas (*Murakami et al., 2012*). The morphology of Txnip-transduced mitochondria in RP cones and uninjected WT cones was examined by transmission electron microscopy (TEM) (*Figure 5—figure supplement 2B–D*). There was an increase in the size of mitochondrial cross sections by Txnip transduction, with a greater increase following transduction with Txnip.C247S in P20 *rd1* cones (*Figure 5A, B*). Mitochondrial membrane potential (ΔΨm), a reflection of mitochondrial ETC function, was also examined using JC-1 dye staining of freshly explanted Txnip-transduced P20 *rd1* retinas (*Reers et al., 1995*). Both Txnip and Txnip.C247S increased the ratio of J-aggregates:JC1-monomers (*Figure 5C, D*), indicating an increased ΔΨm and/or a greater number/size of

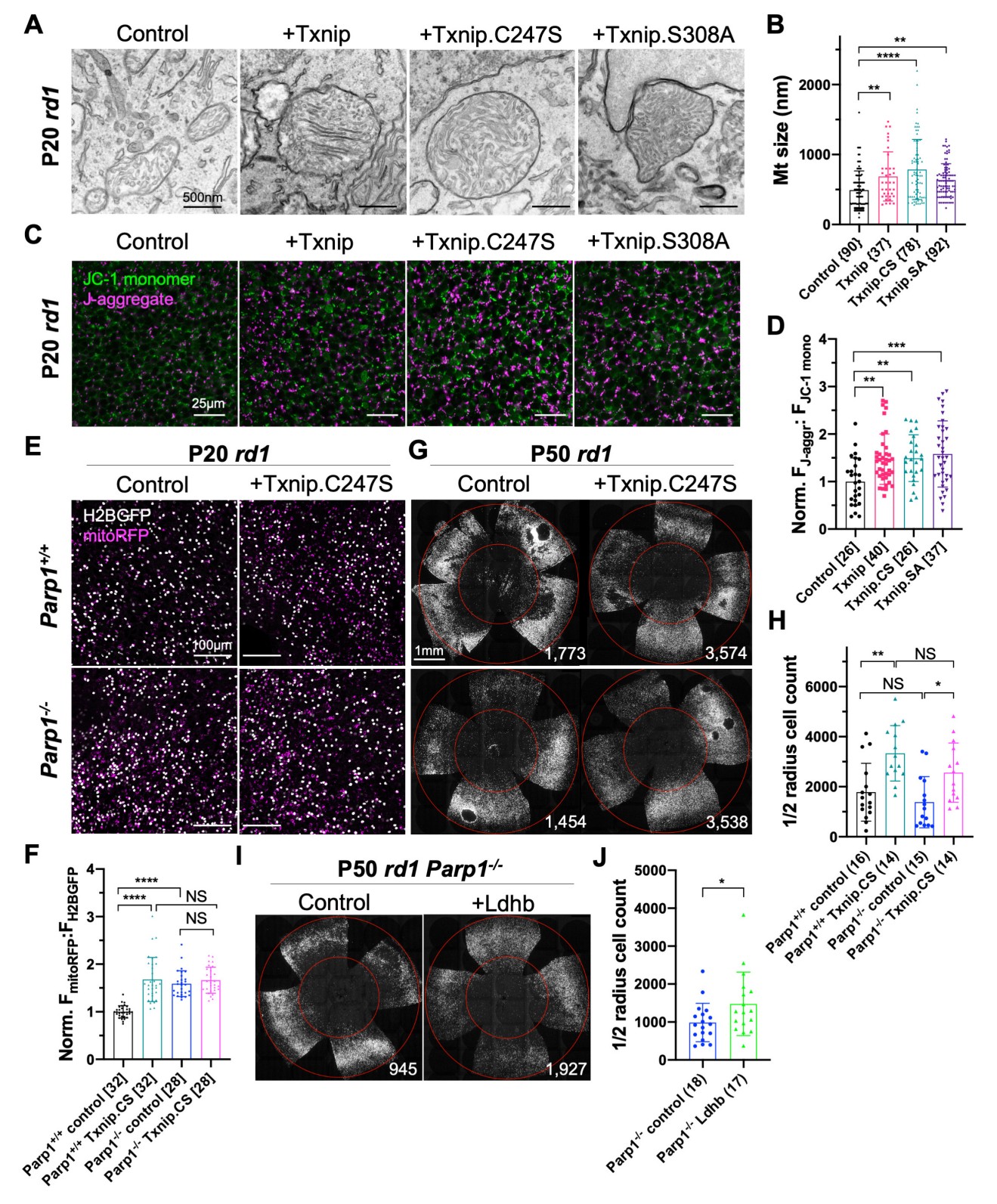

**Figure 5.** Effects of Txnip on retinitis pigmentosa (RP) cone mitochondrial size and function. (**A**) Representative transmission electron microscopy (TEM) images of RP cones from P20 *rd1* cones transduced with wildtype Txnip, Txnip.C247S, or Txnip.S308A (all are AAV8-RedO-Txnip, ≈1 × 10⁹ vg/eye plus AAV8-RedO-H2BGFP, 2.5 × 10⁸ vg/eye), and control (AAV8-RedO-H2BGFP, 2.5 × 10⁸ vg/eye). (**B**) Quantification of mitochondrial diameters from control, Txnip, Txnip.C247S, and Txnip.S308A-transduced cones (same as in **A**). The number in the curly brackets '{}' indicates the sample size, that is,

*Figure 5 continued on next page*

*Figure 5 continued*

the number of mitochondria from multiple cones of $\geq 1$ retina for each condition (five retinas for control, four for Txnip, two for Txnip.C247S, and one for Txnip.S308A). (**C**) Images of JC-1 dye staining (indicator of electron transport chain [ETC] function) in live cones of P20 *rd1* central retina under different conditions (same as in **A**). Magenta: J-aggregate, indicating high ETC function; green: JC-1 monomer, low ETC function, used for normalization. H2BGFP channel, the tracer of AAV infected area, is not shown. (**D**) Quantification of normalized cone JC-1 dye staining (fluorescence intensity of J-aggregate:JC-1 monomer) from live cones in P20 *rd1* retinas (same as in **A**) in different conditions (3–4 images per retina). The number in the square brackets '[]' indicates the sample size, that is, the number of images taken from regions of interest of multiple retinas, in each condition. (**E**) Images of mitoRFP staining (reflecting mitochondrial function) in Txnip.C247S (AAV8-RedO-Txnip.C247S, $1 \times 10^9$ vg/eye, plus AAV8-RedO-H2BGFP, 2.5 $\times 10^8$ vg/eye and AAV8-SynP136-mitoRFP, $1 \times 10^9$ vg/eye) and control (AAV8-RedO-H2BGFP, 2.5 $\times 10^8$ vg/eye plus AAV8-SynP136-mitoRFP, $1 \times 10^9$ vg/eye) cones from fixed P20 *Parp1*$^{+/+}$ *rd1* and *Parp1*$^{-/-}$ *rd1* retinas near the optic nerve head. Magenta: mitoRFP; gray: H2BGFP, for mitoRFP normalization. (**F**) Quantification of normalized mito-RFP:H2BGFP intensity in different conditions (same as in **E**) of P20 *Parp1 rd1* retinas (four images per retina, near optic nerve head). (**G**) Images of P50 *Parp1*$^{+/+}$ *rd1* and *Parp1*$^{-/-}$ *rd1* retinas with H2BGFP (gray)-labeled cones transduced with Txnip. C247S (AAV8-RedO-Txnip, $\approx 1 \times 10^9$ vg/eye plus AAV8-RedO-H2BGFP, 2.5 $\times 10^8$ vg/eye) and control (AAV8-RedO-H2BGFP, 2.5 $\times 10^8$ vg/eye). (**H**) Quantification of H2BGFP-positive cones within the ½ radius of P50 *Parp1*$^{+/+}$ *rd1* and *Parp1*$^{-/-}$ *rd1* retinas transduced with Txnip.C247S or control (same as in **G**). (**I**) Images of P50 *Parp1*$^{-/-}$ *rd1* retinas with H2BGFP (gray)-labeled cones transduced with Ldhb (AAV8-RedO-Ldhb, $1 \times 10^9$ vg/eye, plus AAV8-RedO-H2BGFP, 2.5 $\times 10^8$ vg/eye) or H2BGFP only (AAV8-RedO-H2BGFP, 2.5 $\times 10^8$ vg/eye). (**J**) Quantification of H2BGFP-positive cones within the ½ radius of P50 *Parp1*$^{-/-}$ *rd1* retinas transduced with Ldhb or H2BGFP only (same as in **I**). Error bar: standard deviation. NS: not significant, p>0.05, *p<0.05, **p<0.01, ***p<0.001, **** p< or <<0.0001. Txnip.CS: Txnip.C247S; Txnip.SA: Txnip.S308A. RedO: red opsin promoter; AAV: adeno-associated virus; Ldhb: lactate dehydrogenase b.

The online version of this article includes the following source data and figure supplement(s) for figure 5:

**Source data 1.** Differentially expressed genes in cones infected by AAV8-RedO-Txnip ($1 \times 10^9$ vg/eye plus AAV8-SynP136-H2BGFP, $1 \times 10^9$ vg/eye) vs. control (AAV8-SynP136-H2BGFP, $1 \times 10^9$ vg/eye) in common between two retinitis pigmentosa strains (*rd1* and *Rho*$^{-/-}$). RedO: red opsin promoter; AAV: adeno-associated virus.

**Source data 2.** Differentially expressed gene(s) in cones infected by AAV8-RedO-Txnip ($1 \times 10^9$ vg/eye plus AAV8-SynP136-H2BGFP, $1 \times 10^9$ vg/eye) vs. control (AAV8-SynP136-H2BGFP, $1 \times 10^9$ vg/eye) in common between two wildtype strains (BALB/c and C57BL6/J). RedO: red opsin promoter; AAV: adeno-associated virus.

**Source data 3.** Cone *Txnip* mRNA raw reads in the RNA-seq data (from 1000 FACS cones per retina).

**Source data 4.** Cone mRNA raw reads from RNA-seq of all 35 retinas used in the study.

**Figure supplement 1.** Additional figures for RNAseq results of Txnip and effects of Txnip on retinitis pigmentosa (RP) cone mitochondria.

**Figure supplement 2.** Additional figures for effects of Txnip on retinitis pigmentosa (RP) cone mitochondria.

mitochondria with a high ΔΨm following Txnip overexpression. This finding was further investigated in vivo using infection by an AAV encoding mitoRFP, which only accumulates in mitochondria with a high ΔΨm (*Brodier et al., 2020*; *Hood et al., 2003*). Compared to the control cones without Txnip transduction, the intensity of mitoRFP was higher in P20 *rd1* cones transduced with Txnip (*Figure 5— figure supplement 1C, D*).

A previous study identified 15 proteins that interact with Txnip.C247S (*Forred et al., 2016*). Among these interactors was Parp1, which can negatively affect mitochondria through deleterious effects on the mitochondrial genome (*Hocsak et al., 2017*; *Szczesny et al., 2014*), as well as effects on inflammation and other cellular pathways (*Fehr et al., 2020*). Due to the similarities between the effects of Txnip addition and of Parp1 inhibition on mitochondria, Parp1 was tested for a potential role in Txnip-mediated rescue. Parp1 expression was first examined by IHC and found to be enriched in cone inner segments, which are packed with mitochondria (*Hoang et al., 2002*), as well as in cone nuclei (*Figure 5—figure supplement 1G*). Interestingly, when a GFP-Txnip fusion protein was expressed in cones, it also was found in these regions (*Figure 1—figure supplement 1B*). To test for a role of Parp1, *Parp1*$^{-/-}$ mice were bred to *rd1* mice, and their cone mitochondria were examined by TEM and mitoRFP. *Parp1*$^{-/-}$ *rd1* cones possessed larger mitochondria (*Figure 5—figure supplement 1H, I*) and higher mitoRFP signals than cones from *Parp1*$^{+/+}$ *rd1* controls (*Figure 5E, F*). Addition of Txnip.C247S to *Parp1*$^{-/-}$ *rd1* cones did not alter the mitoRFP signals (*Figure 5E, F*). When Txnip.C247S was added to *Parp1*$^{-/-}$ *rd1* retinas, cone survival was similar to that of Txnip. C247S-transduced *Parp1*$^{+/+}$ *rd1* retinas, showing that Txnip-mediated survival does not require Parp1 (*Figure 5G, H*). *Parp1*$^{-/-}$ *rd1* cone survival also was similar to the *Parp1*$^{+/+}$ *rd1* cones (*Figure 5G, H*). The *rd1* cone degeneration seemed to be faster in these *Parp1*$^{+/+}$ and *Parp1*$^{-/-}$ mice (on 129S background) for unknown reason(s).

The discordance between improved mitochondria and cone survival in these experiments suggested that mitochondrial improvement alone is not sufficient to prolong cone survival. This is

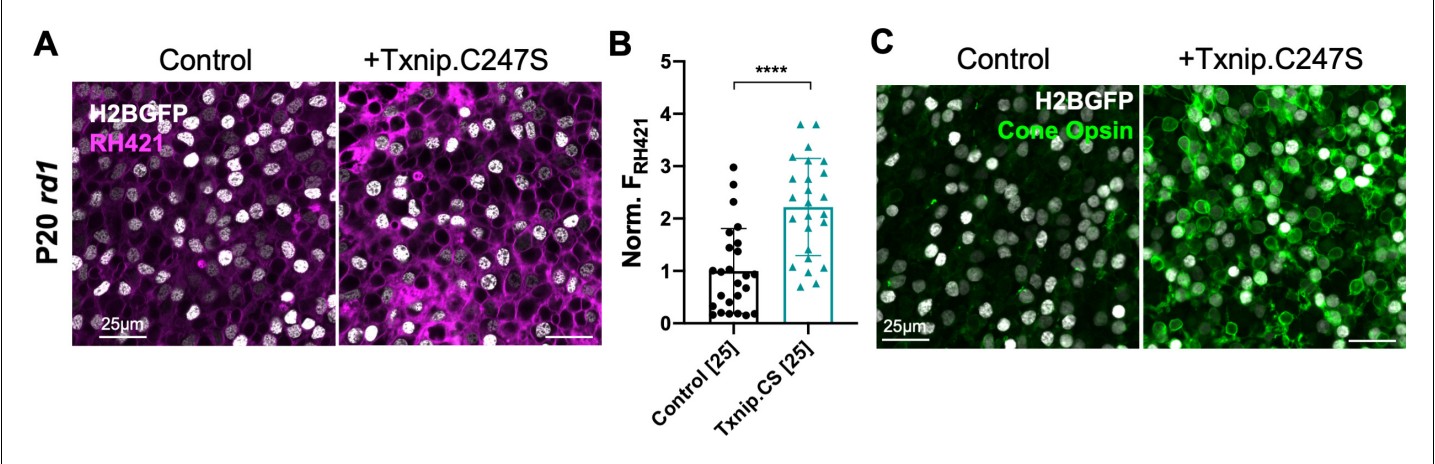

**Figure 6.** Effect of Txnip on Na$^+$/K$^+$ ATPase pump function and cone opsin expression in retinitis pigmentosa cones. (A) Images of live ex vivo RH421 stained cones in P20 *rd1* retinas transduced with Txnip.C247S (AAV8-RedO-Txnip C247S, ≈1 × 10$^9$ vg/eye plus AAV8-RedO-H2BGFP, 2.5 × 10$^8$ vg/eye) and control (AAV8-RedO-H2BGFP, 2.5 × 10$^8$ vg/eye) and cultured in lactate-only medium. Magenta: RH421 fluorescence, proportional to Na$^+$/K$^+$ ATPase function; gray: H2BGFP, tracer of infection. (B) Quantification of normalized RH421 fluorescence intensity from Txnip.C247S-transduced cones relative to control in P20 *rd1* retinas cultured in lactate-only medium (same as in A, five images per retina). The number in the square brackets '[]' indicates the sample size, that is, the number of images taken from regions of interest of multiple retinas, in each condition. Txnip.CS: Txnip.C247S. (C) Immunohistochemistry with anti-s-opsin plus anti-m-opsin antibodies in the center of P50 *rd1* retinas transduced with Txnip.C247S or control. Green: cone-opsins; gray: H2BGFP, tracer of infection. Error bar: standard deviation. **** p< or <<0.0001. RedO: red opsin promoter; AAV: adeno-associated virus.

consistent with the observations from transduction with Txnip.S308A, as well as Txnip + siLdhb, both of which failed to prolong *rd1* cone survival despite improvements in mitochondria (*Figure 2A, B*, *Figure 5A–D*, *Figure 5—figure supplement 1C–F*). To test if improved cone survival requires enhanced lactate catabolism in addition to mitochondrial improvement, we delivered Ldhb to *Parp1$^{-/-}$ rd1* cones. Unlike on *Parp1$^{+/+}$* background (*Figure 8—figure supplement 1D*), a small but significant improvement in cone survival was observed (*Figure 5I, J*).

## Txnip enhances Na$^+$/K$^+$ pump function and cone opsin expression

The results above suggest that Txnip may prolong RP cone survival by enhancing lactate catabolism via Ldhb, which may lead to greater ATP production by the OXPHOS pathway. Cone photoreceptors are known to require high levels of ATP to maintain their membrane potential, relying primarily upon the Na$^+$/K$^+$ ATPase pump (*Ingram et al., 2020*). To investigate whether Txnip affects the function of the Na$^+$/K$^+$ pump in RP cones, freshly explanted P20 *rd1* retinas were stained with RH421, a fluorescent small-molecule probe for Na$^+$/K$^+$ pump function (*Fedosova et al., 1995*). Addition of Txnip improved Na$^+$/K$^+$ pump function of these cones in lactate medium as reflected by an increase in RH421 fluorescence (*Figure 6A, B*), consistent with Txnip enabling greater utilization of lactate as fuel. In RP cones, it is also known that protein expression of cone opsin is downregulated, postulated to be due to insufficient energy supply (*Punzo et al., 2009*). Compared to control, greater anti-opsin staining was observed in Txnip-transduced *rd1* cones at P50 (*Figure 6C*), further supporting the idea that Txnip improves the energy supply to RP cones.

## Dominant-negative HIF1α improves RP cone survival

If improved lactate catabolism and OXPHOS are at least part of the mechanism of Txnip rescue, RP cone survival might be promoted by other molecules serving similar functions. HIF1α can upregulate the transcription of glycolytic genes (*Majmundar et al., 2010*). Increased glycolytic enzyme levels might push RP cones to further rely on glucose, rather than lactate, to their detriment if glucose is limited. To investigate whether HIF1α might play a role in cone survival, a WT and a dominant-negative HIF1α (dnHIF1α) allele (*Jiang et al., 1996*) were delivered to *rd1* retinas using AAV. A target gene of HIF1α, *Vegfa*, which might improve blood flow and thus nutrient delivery (but might also

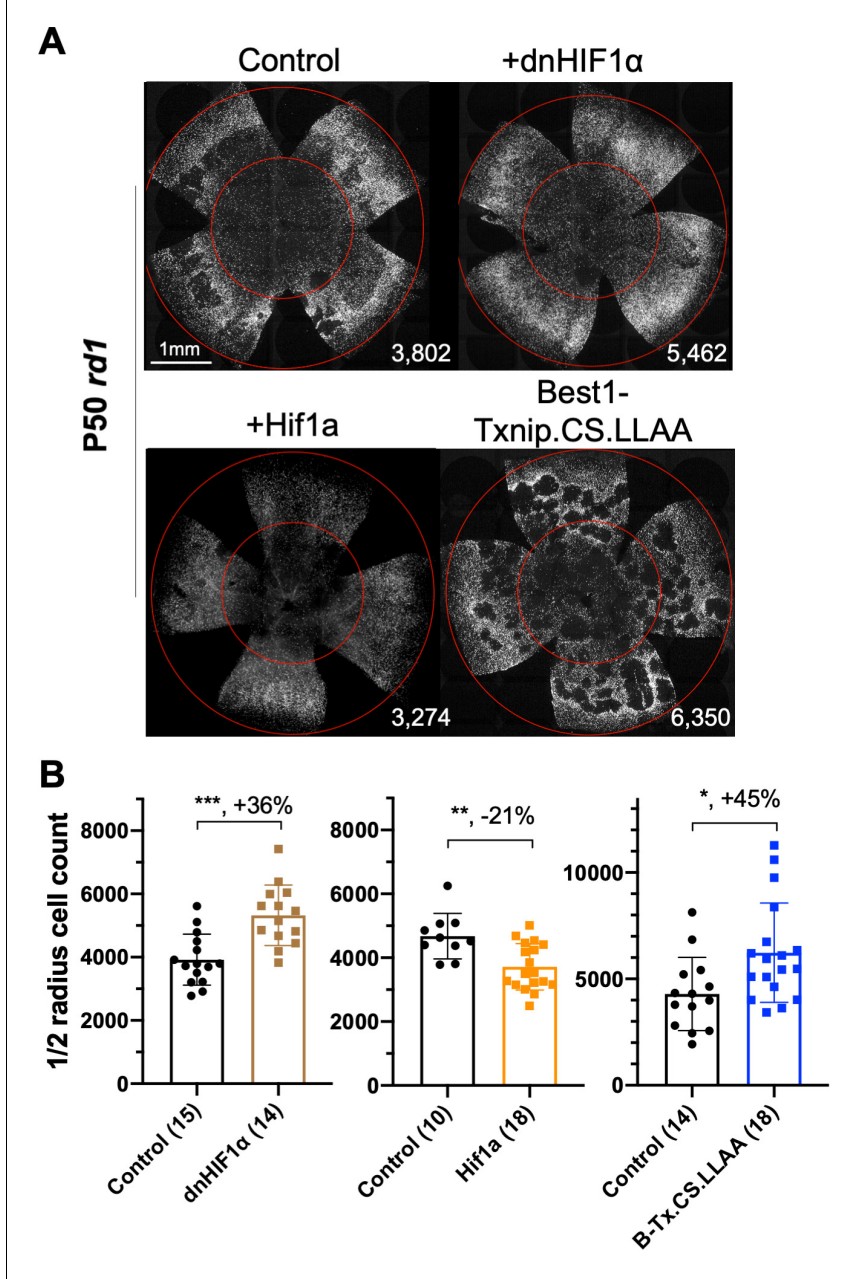

**Figure 7.** Effect of dominant negative HIF1α and Best1-Txnip.C247S.LL351 and 352AA on retinitis pigmentosa cone survival. (**A**) Images of P50 *rd1* retinas with H2BGFP (gray)-labeled cones transduced with dnHIF1α (AAV8-RO1.7-dnHIF1α, $1 \times 10^9$ vg/eye), Hif1a (AAV8-SynPVI-Hif1a, SynPVI is an alternative cone-specific promoter, $1 \times 10^9$ vg/eye), Best1-Txnip.C247S.LL351 and 352AA (Txnip.CS.LLAA, driven by a retinal pigmented epithelium-specific promoter; AAV8, $5 \times 10^8$ vg/eye; all including AAV8-RedO-H2BGFP, $2.5 \times 10^8$ vg/eye), or control (AAV8-RedO-H2BGFP, $2.5 \times 10^8$ vg/eye). (**B**) Quantification of H2BGFP-positive cones within the ½ radius of P50 *rd1* retinas transduced with dnHIF1α, Hif1a, Best1-Txnip.C247S.LL351, and 352AA or control (same as in **A**). B-Tx.CS.LLAA: Best1-Txnip.C247S.LL351 and 352AA. Error bar: standard deviation. *p<0.05, **p<0.01, ***p<0.001. RedO: red opsin promoter; AAV: adeno-associated virus.

The online version of this article includes the following figure supplement(s) for figure 7:

**Figure supplement 1.** Additional figures for various vectors effect on retinitis pigmentosa (RP) cone survival.

increase inflammation), also was tested. The dnHIF1α increased *rd1* cone survival, while WT Hif1a

and Vegf each decreased cone survival (*Figure 7*, *Figure 7—figure supplement 1D, E*).

## Txnip effects on GLUT1 levels in the RPE and cone survival

To determine if retention of glucose by the RPE might underlie a glucose shortage for cones (*Kanow et al., 2017*; *Wang et al., 2016*), we attempted to reprogram RPE metabolism to a more 'OXPHOS' and less 'glycolytic' status by overexpressing Txnip or dnHIF1α with an RPE-specific promoter, the Best1 promoter (*Esumi et al., 2009*). The goal was to increase lactate consumption in the RPE, thus freeing up more glucose for delivery to cones. However, no RP cone rescue was observed (*Figure 7—figure supplement 1B*), possibly due to a clearance of GLUT1 by Txnip from the surface of RPE cells, which would create a glucose shortage for both the RPE and the cones (*Swarup et al., 2019*; *Figure 7—figure supplement 1A*). To examine the level of GLUT1 in the RPE following introduction of WT Txnip, or Txnip.C247S.LL351 and 352AA, which should prevent efficient removal of GLUT1 (see background in previous section), IHC for GLUT1 was carried out. This assay showed that AAV-Best1-Txnip.LL351 and 352AA did result in less clearance of GLUT1 from the surface of the RPE relative to WT Txnip (*Figure 7—figure supplement 1A*). Best1-Txnip.C247S.LL351 and 352AA was then tested for *rd1* cone rescue, where it was found to improve cone survival (*Figure 7*), in keeping with the model that the RPE retains glucose to the detriment of cones in RP.

## Combination of Txnip.C247S with other rescue genes provides an additive effect

Finally, as our goal is to provide effective, generic gene therapy for RP, and potentially other diseases that affect photoreceptor survival, we used combinations of AAVs that encode genes that we have previously shown prolong RP cone survival and vision. The combination of Txnip.C247S expression in cones, with expression of Nrf2, a gene with anti-oxidative damage and anti-inflammatory activity, in the RPE, provided an additive effect on cone survival relative to either gene alone (*Figure 8A, B*). This combination also preserved a structure resembling cone outer segments. In WT cones, opsin protein is localized to the outer segment, where photon detection and phototransduction take place. During degeneration in RP, the cone outer segments collapse, and opsin is mislocalized to the plasma membrane (*Figure 8C*, *Figure 8—figure supplement 1A*). The combination of RedO-Txnip.C247S and Best1-Nrf2 led to the localization of opsin protein to the outer segment-like structure, rather than to the plasma membrane. An additional morphological phenotype that is especially prominent in the FVB *rd1* strain is that of 'craters' in the photoreceptor layer. These are circumscribed areas without cones that are obvious when the retina is viewed as a flat-mount. AAV-Best1-Nrf2 alone suppressed the formation of these craters (*Figure 8A*; *Wu et al., 2021*), while AAV-RedO-Txnip did not, despite the fact that AAV-RedO-Txnip.C247S provided the most robust RP cone rescue (*Figure 2A*, *Figure 8A, D*). It was also noted that dnHIF1α decreased, and Best1-Txnip.C247S.LL351 and 352AA increased, the FVB-specific retinal craters, while both vectors prolonged RP cone survival (*Figure 7A*). The significance of these craters is thus unclear, as is the mechanism of their formation, though these data point to an origin within the RPE.

An additional combination that was tested was AAV-RedO-Txnip.C247S with AAV-RedO-Tgfb1, an anti-inflammatory gene that our previous studies showed could eliminate the craters on its own (*Wang et al., 2020*). This combination did not improve cone survival beyond that of Txnip alone, but almost completely eliminated the craters in an additive fashion with Txnip (*Figure 8D, E*). In addition, other genes (*Hk2, Ldhb,* and *Cx3cl1*) and Nrf2 were expressed specifically in cones in combination with WT Txnip, but did not provide any obvious improvement over Txnip alone (*Figure 8—figure supplement 1*).

## Discussion

Photoreceptors have been characterized as being highly glycolytic, even under aerobic conditions, as originally described by *Warburg, 1925*. Glucose appears to be supplied primarily from the circulation, via the RPE, which has a high level of GLUT1 (*Gospe et al., 2010*). Photoreceptors, at least rods, carry out glycolysis to support anabolism, to replace their outer segments (*Chinchore et al., 2017*), and contribute ATP, to run their ion pumps (*Okawa et al., 2008*). If glucose becomes limited, as has been proposed to occur in RP, cones may have insufficient fuel for their needs. To explore whether we could develop a therapy to address some of these metabolic shortcomings in RP, we

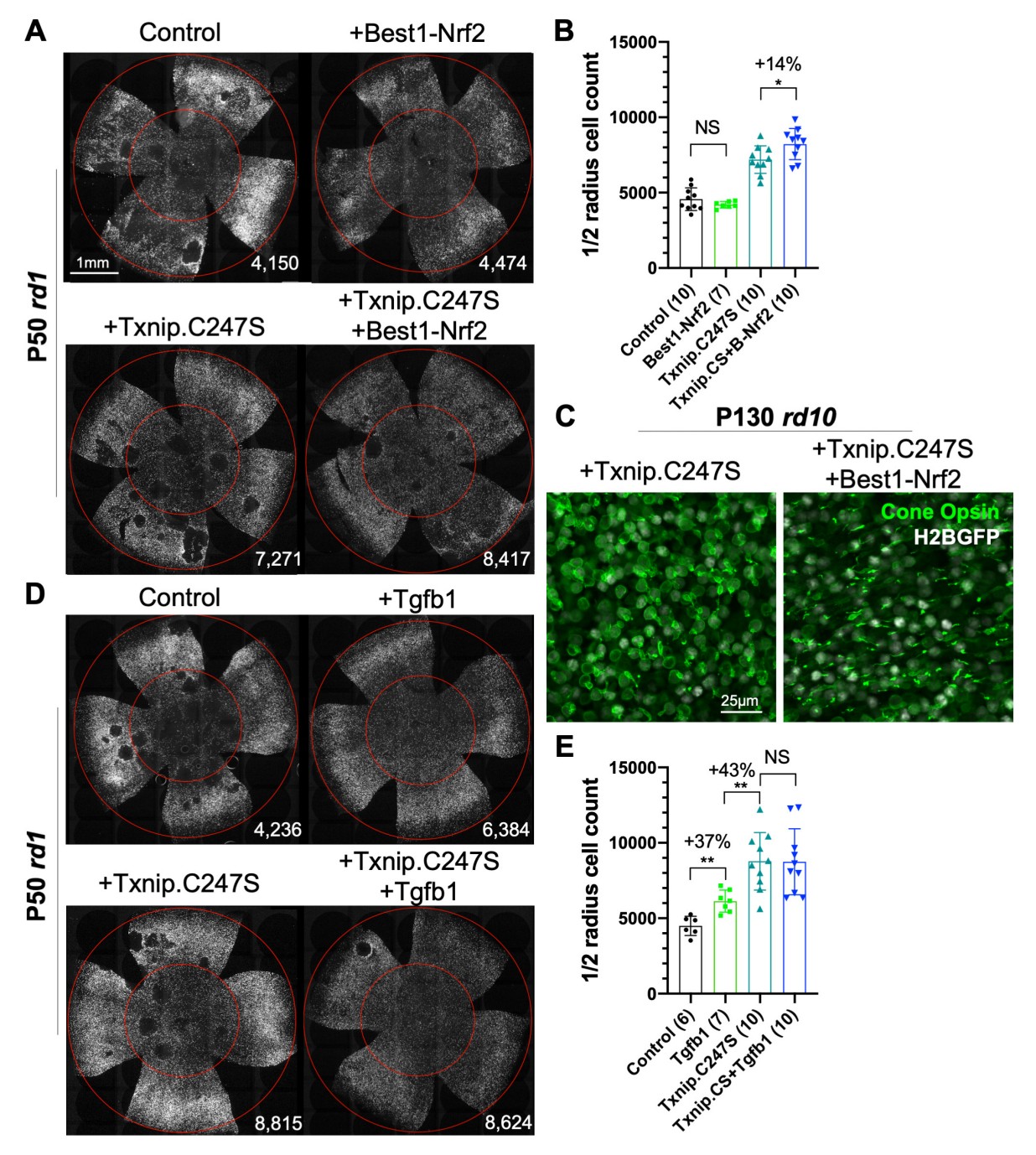

**Figure 8.** Effect of combinations of Txnip.C247S with Best1-Nrf2 or Tgfb1 on retinitis pigmentosa cone survival. (**A**) Images of P50 rd1 retinas with H2BGFP (gray)-labeled cones transduced with Nrf2 (AAV8-Best1-Nrf2, $2.5 \times 10^8$ vg/eye), Txnip.C247S (AAV8-RedO-Txnip.C247S, $1 \times 10^9$ vg/eye), Txnip. C247S (AAV8-RedO-Txnip.C247S, $1 \times 10^9$ vg/eye) + Best1-Nrf2 (AAV8-Best1-Nrf2, $2.5 \times 10^8$ vg/eye), or control (AAV8-RedO-H2BGFP, $2.5 \times 10^8$ vg/ eye). All experimental vector injections included AAV8-RedO-H2BGFP, $2.5 \times 10^8$ vg/eye. (**B**) Quantification of H2BGFP-positive cones within the ½ radius of P50 *rd1* retinas transduced with Best1-Nrf2, Txnip.C247S, Txnip.C247S + Best1-Nrf2, or control. Txnip.CS: Txnip.C247S. B-Nrf2: Best1-Nrf2. (**C**) Immunohistochemistry with anti-S-opsin plus anti-M-opsin antibodies in the center of P130 *rd10* retinas transduced with Txnip.C247S (left panel) or Txnip.C247S + Best1-Nrf2 (right panel). Green: cone-opsins; gray: H2BGFP, tracer of infection. (**D**) Images of P50 rd1 retinas with H2BGFP (gray)-labeled cones transduced with Tgfb1 (AAV8-RedO, $1 \times 10^9$ vg/eye), Txnip.C247S (AAV8-RedO-Txnip.C247S, $1 \times 10^9$ vg/eye), Txnip.C247S (AAV8-RedO-Txnip. C247S, $1 \times 10^9$ vg/eye) + Tgfb1 (AAV8-RedO-Tgfb1, $1 \times 10^9$ vg/eye), or control (AAV8-RedO-H2BGFP, $2.5 \times 10^8$ vg/eye). All experimental vector injections included AAV8-RedO-H2BGFP, $2.5 \times 10^8$ vg/eye. (**E**) Quantification of H2BGFP-labeled cones within the ½ radius of P50 *rd1* retinas

*Figure 8 continued on next page*

Figure 8 continued

transduced with control, Tgfb1, Txnip.C247S, or Txnip.C247S + Tgfb1. Error bar: standard deviation. NS: not significant, p>0.05, *p<0.05, **p<0.01. RedO: red opsin promoter; AAV: adeno-associated virus.

The online version of this article includes the following figure supplement(s) for figure 8:

**Figure supplement 1** Effect of combinations of Txnip with various vectors on retinitis pigmentosa (RP) cone survival.

delivered many different types of genes that might alter metabolic programming. From these, Txnip had the strongest improvement on cone survival and vision (*Figure 1*, *Figure 1—figure supplements 1* and *2*). This was surprising as Txnip has been shown to inhibit glucose uptake by binding to and aiding in the removal of GLUT1 from plasma membrane (*Wu et al., 2013*). Moreover, it inhibits the anti-oxidation proteins, the thioredoxins, again by direct binding (*Junn et al., 2000*; *Nishinaka et al., 2001*; *Nishiyama et al., 1999*), so would have been predicted to increase oxidative damage and therefore decrease cone survival. The results with Txnip in its WT form, and from the study of several mutant alleles, provide some insight into how it might benefit cones. The Txnip. C247S allele prevents binding to thioredoxins and gave enhanced cone survival relative to WT Txnip (*Figure 2*, *Figure 2—figure supplement 1*). We speculate that, by being free of this interaction, the C247S mutant protein may be more available for other Txnip-mediated activities. In addition, thioredoxin may be made more available for its role in fighting oxidative damage following transduction of the C247S mutant, rather than the WT Txnip allele.

The mechanisms by which Txnip might benefit cones are not fully known, but a study of Txnip's function in skeletal muscle suggested that it plays a role in fuel selection (*DeBalsi et al., 2014*). If glucose is limited in RP, then cones may need to switch from a reliance on glucose and glycolysis to an alternative fuel(s), such as ketones, fatty acids, amino acids, or lactate. Cones express *oOxct1* mRNA (*Shekhar et al., 2016*), which encodes a critical enzyme for ketone catabolism, suggesting that cones are capable of ketolysis. In addition, a previous study showed that lipids might be an alternative energy source for cones by β-oxidation (*Joyal et al., 2016*). It is likely that cones can use these alternative fuels to meet their intense energy demands (*Ingram et al., 2020*; *Figure 6*). However, the Txnip rescue did not depend on ketolysis or β-oxidation (*Figure 3*). Due to the diversity of amino acid catabolic pathways, we did not study whether these pathways were required for Txnip's rescue effect. However, we did discover that Ldhb, which converts lactate to pyruvate, was required. This is an interesting switch as photoreceptors normally have high levels of Ldha (*Figure 5—source data 4*) and produce lactate (*Chinchore et al., 2017*; *Kanow et al., 2017*). An important factor in the reliance on Ldhb could be the availability of lactate, which is highly available from serum (*Hui et al., 2017*). Lactate could be transported via the RPE and/or Müller glia, and/or the internal retinal vasculature that comes in closer proximity to cones after rod death. Ketones are usually only available during fasting, and lipids are hydrophobic molecules that are slow to be transported across the plasma membranes. Moreover, lipids are required to rebuild the membrane-rich outer segments, and thus might be somewhat limited. Ldhb is not sufficient, however, to delay RP cone degeneration as its overexpression did not promote RP cone survival.

Txnip-transduced RP cones also had larger mitochondria with a greater membrane potential and likely were able to use the pyruvate produced by Ldhb for greater ATP production via OXPHOS. Indeed, Txnip-transduced cones had an enhanced ATP:ADP ratio (*Figure 4*). However, healthier mitochondria were not sufficient to prolong RP cone survival. Txnip.S308A led to larger mitochondria than control mitochondria as seen by TEM, as well as brighter JC-1 staining and mitoRFP signals, which are indicators of better mitochondrial health, but this allele did not induce greater cone survival (*Figure 5*, *Figure 5—figure supplement 1*). Moreover, as Txnip has been shown to interact with Parp1, which can negatively affect mitochondria, we investigated if Parp1 knockout mice might have cones that survive longer in RP. Indeed, the Parp1 knockout mitochondria appeared to be healthier, but Parp1 knockout retinas did not have better RP cone survival than Parp1-WT *rd1* retinas (*Figure 5*, *Figure 5—figure supplement 1*). In addition, cone rescue by Txnip was not changed in the Parp1 knockout retinas (*Figure 5*).

The well-described effects of Txnip on the removal of GLUT1 from the cell membrane might seem at odds with the promotion of cone survival. It could be that removal of GLUT1 from the plasma membrane of cones forces the cones to choose an alternative fuel, such as lactate, and

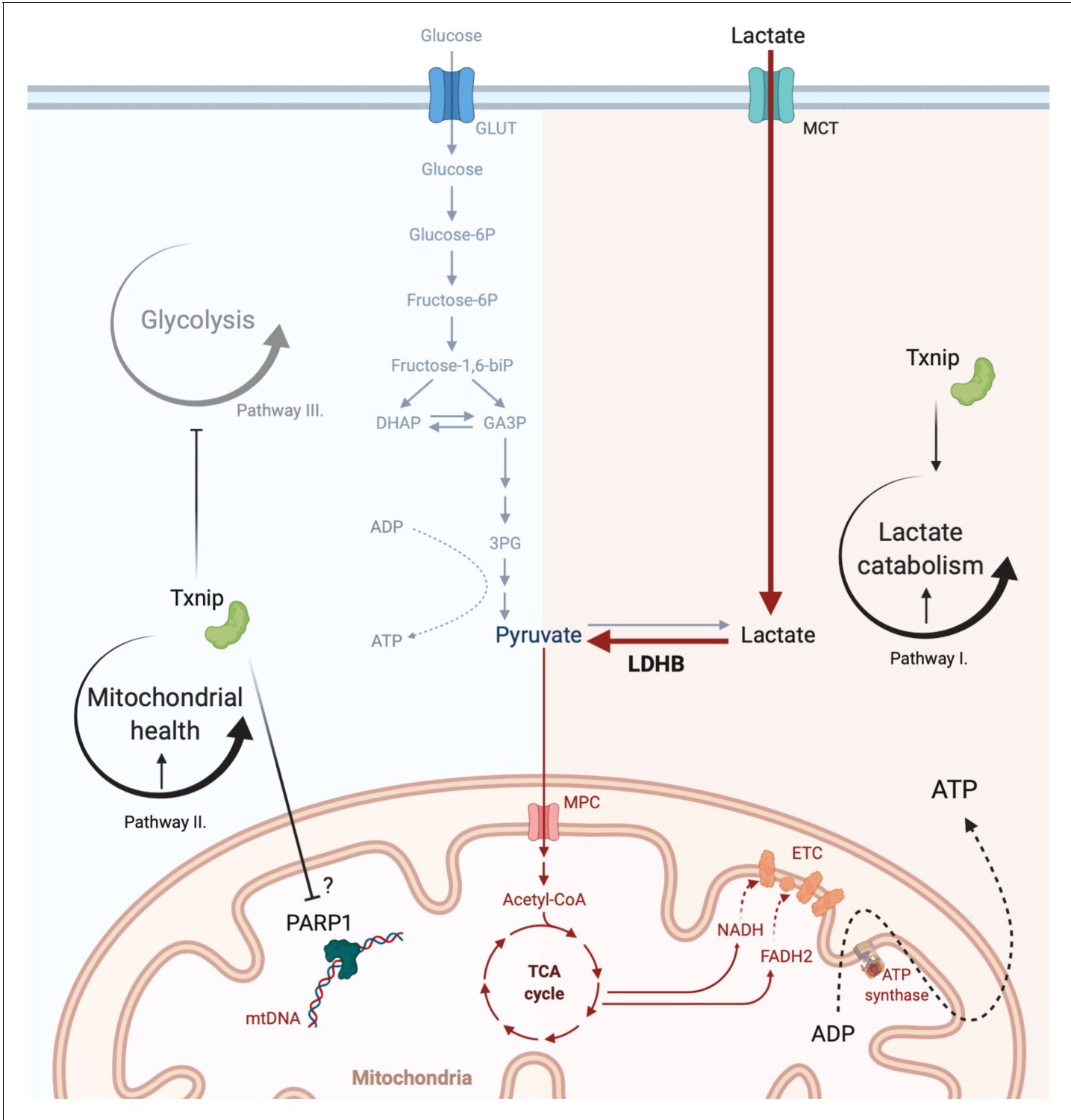

**Figure 9.** Model for the mechanism of Txnip-mediated cone rescue. The data suggest that at least two pathways are required for Txnip rescue – Pathway I: enhancement of lactate catabolism, which requires the function of LDHB; and pathway II: improved mitochondrial health, possibly through mitochondrial Parp1 inhibition. An additional pathway, pathway III: inhibiting glycolysis by removing glucose transporter 1 (GLUT1) from cell surface, may partially contribute to rescue by working with improved lactate metabolism and improved mitochondrial health, and/or unidentified pathways. This diagram was created from a BioRender Template. MCT: monocarboxylate transporter; 3PG: 3 phosphoglyceric acid; TCA: tricarboxylic acid cycle.

perhaps others too. Interestingly, as GLUT1 knockdown was not sufficient for cone survival, Txnip must not only lead to a reduction in membrane-localized GLUT1, but also potentiate a fuel switch, via an unknown mechanism(s) that at least involves an increase of Ldhb activity. A reduction in glycolysis might also lead to a fuel switch. Introduction of dnHIF1α, which should reduce expression of glycolytic enzymes, also benefited cones, while introduction of WT HIF1α did not (*Figure 7*). HIF1α has many target genes and may alter pathways in addition to that of glycolysis, thus also potentiating a fuel switch once glycolysis is downregulated. An additional finding supporting the notion that the level of glycolysis is important for cone survival was the observation that AAV-Pfkm plus AAV-Hk1 led to a reduction in cone survival (*Figure 1—figure supplement 1E*). Phosphorylation of glucose by Hk1 followed by phosphorylation of fructose-6-phosphate by the Pfkm complex commits glucose to glycolysis at the cost of ATP. These AAVs may have promoted the flux of glucose through glycolysis, which may have inhibited a fuel switch, and/or depleted the ATP pool, for example, if downstream glycolytic intermediates were used for anabolic needs so that ATP production by glycolysis did not occur, and pyruvate was not produced for conversion to acetyl CoA and ATP production by the mitochondria.

The observations described above suggest that at least two different pathways are required for the promotion of cone survival by Txnip (*Figure 9*). One pathway requires lactate utilization via Ldhb, but as Ldhb was not sufficient, another pathway is also required. As greater mitochondrial health was observed following Txnip transduction, a second pathway may include the effects on mitochondria. This notion is supported by the observation that the addition of Ldhb to *Parp1*⁻/⁻ *rd1* cones, which have healthier mitochondria, led to improved cone survival (*Figure 5*). Txnip alone may be able to promote cone health by impacting both lactate catabolism and mitochondrial health. There may be additional pathways required as well.

The effects of Txnip alleles expressed only in the RPE provide some support for the hypothesis that the RPE transports glucose to cones for their use, while primarily using lactate for its own needs (*Kanow et al., 2017*; *Swarup et al., 2019*). Lactate is normally produced at high levels by photoreceptors in the healthy retina. When rods, which are 97% of the photoreceptors, die, lactate production goes down dramatically. The RPE might then need to retain glucose for its own needs. Introduction of an allele of Txnip, C247S.LL351 and 352AA, to the RPE provided a rescue effect for cones, while introduction of the WT allele of Txnip to the RPE did not. The LL351 and 352AA mutations lead to a loss of efficiency of the removal of GLUT1 from the plasma membrane, while the C247S mutation might create a less glycolytic RPE. The combination of these mutations might then allow more glucose to flow to cones. The untreated RP cones seem to be able to use glucose at a high concentration for ATP production, at least in freshly explanted retinas (*Figure 4A*). These findings are also consistent with the reported mechanism for cone survival promoted by RdCVF, a factor that is proposed to improve glucose uptake by RP cones, which might be important if glucose is present in low concentration due to retention by the RPE (*Aït-Ali et al., 2015*; *Byrne et al., 2015*).

As cones face multiple challenges in the degenerating RP retina, we tested Txnip in combination with genes that we have found to promote cone survival via other mechanisms. The combination of Txnip with vectors fighting oxidative stress and inflammation (AAV-Best1-Nrf2) supported greater cone survival than any of these treatments alone. These combinations utilize cell type-specific promoters, reducing the chances of side effects from global expression of these genes. Of note, the Nrf2 expression was limited to the RPE, which we recently showed leads to much improved RPE survival and morphology in RP mice (*Wu et al., 2021*). Best1-Nrf2 was additive to RedO-Txnip for cone survival. This finding is in keeping with the interdependence of photoreceptors and the RPE, which is undoubtedly important not only in a healthy retina, but in disease as well.

## Materials and methods

**Key resources table**

| Reagent type (species) or resource | Designation | Source or reference | Identifiers | Additional information |
|---|---|---|---|---|

*Continued on next page*

*Continued*

| Reagent type (species) or resource | Designation | Source or reference | Identifiers | Additional information |
|---|---|---|---|---|
| Genetic reagent (*Mus musculus*) | *Pde6b*<sup>rd1</sup> | Charles River; Taconic | Stock #: 207; FVB/NTac. MGI: 1856373 | |
| Genetic reagent (*M. musculus*) | *Pde6b*<sup>rd10</sup> | Jackson Laboratory | Stock #: 004297 MGI:2388259 | |
| Genetic reagent (*M. musculus*) | *Rho*<sup>-/-</sup> | Janis Lem (Tufts University) | MGI:2680822 | PMID:9892703 |
| Genetic reagent (*M. musculus*) | *Parp1*<sup>-/-</sup> | Jackson Laboratory | Stock #: 002779. MGI:1857862 | |
| Antibody | Rabbit anti-GLUT1 | Alpha Diagnostics | GT11-A | IHC (1:300) |
| Antibody | Rabbit anti-PARP1 | Abcam | ab227244 | IHC (1:300) |
| Antibody | Chicken anti-GFP | Abcam | ab13970 | IHC (1:1000) |
| Antibody | Goat anti-FLAG | Abcam | ab1257 | IHC (1:2000) |
| Antibody | Rabbit anti-ARR3 | Millipore Sigma | AB15282 | IHC (1:1000) |
| Antibody | Rabbit anti-OPN1MW | Millipore Sigma | AB5405 | IHC (1:600) |
| Antibody | Rabbit anti-OPN1SW | Millipore Sigma | AB5407 | IHC (1:200) |
| Genetic reagent (*M. musculus*) | Txnip cDNA | GeneCopoeia | Cat. #: Mm07552 NCBI: NM_001009935.2 | |
| Genetic reagent (*M. musculus*) | Hif1a cDNA | GeneCopoeia | Cat. #: Mm30422 NCBI: NM_010431.2 | |
| Genetic reagent (*M. musculus*) | Hk2 cDNA | GeneCopoeia | Cat. #: Mm03044 NCBI: NM_013820.3 | |
| Genetic reagent (*M. musculus*) | Ldha cDNA | GeneCopoeia | Cat. #: Mm28710 NCBI: NM_001136069.2 | |
| Genetic reagent (*M. musculus*) | Ldhb cDNA | GeneCopoeia | Cat. #: Mm03608 NCBI: NM_008492.2 | |
| Genetic reagent (*M. musculus*) | Slc2a1 cDNA | GeneCopoeia | Cat. #: Mm21137 NCBI: NM_011400.3 | |
| Genetic reagent (*M. musculus*) | Bsg1 cDNA | GeneCopoeia | Cat. #: Mm01471 NCBI: NM_009768.2 | |
| Genetic reagent (*M. musculus*) | Cpt1a cDNA | GeneCopoeia | Cat. #: Mm20470 NCBI: NM_013495.2 | |
| Genetic reagent (*M. musculus*) | Oxct1 cDNA | GeneCopoeia | Cat. #: Mm08941 NCBI: NM_024188.6 | |
| Genetic reagent (*M. musculus*) | Mpc1 cDNA | GeneCopoeia | Cat. #: Mm41054 NCBI: NM_001364919.1 | |
| Genetic reagent (*M. musculus*) | Mpc2 cDNA | GeneCopoeia | Cat. #: Mm19410 NCBI: BC018324.1 | |
| Genetic reagent (*Homo sapiens*) | Nrf2 cDNA | GeneCopoeia | Cat. #: T3128 NCBI: NM_006164.4 | |
| Genetic reagent (*H. sapiens*) | Hk1 | William Hahn and David Root (via Addgene) | Cat. #: 23730 | PMID:21107320 |
| Genetic reagent (*H. sapiens*) | Pfkm | William Hahn and David Root (via Addgene) | Cat. #: 23728 | PMID:21107320 |
| Genetic reagent (*H. sapiens*) | Pkm2 | William Hahn and David Root (via Addgene) | Cat. #: 23757 | PMID:21107320 |
| Genetic reagent (*H. sapiens*) | Pkm1 | Lewis Cantley and Matthew Vander Heiden (via Addgene) | Cat. #: 44241 | PMID:18337815 |
| Recombinant DNA reagent | DsRed2-mito | Addgene (Michael Davidson) | 55838 | |

*Continued on next page*

*Continued*

| Reagent type (species) or resource | Designation | Source or reference | Identifiers | Additional information |
|---|---|---|---|---|
| Recombinant DNA reagent | GFP-Txnip | Addgene (Clark Distelhorst) | 18758 | PMID:16301999 |
| Recombinant DNA reagent | PercevalHR | Addgene (Gary Yellen) | 49082 | PMID:24096541 |
| Recombinant DNA reagent | pAAV-RedO | Botond Roska | | PMID:20576849 |
| Recombinant DNA reagent | pAAV-SynPVI (ProA7) | Botond Roska | | PMID:31285614 |
| Recombinant DNA reagent | pAAV-SynP136 (ProA1) | Botond Roska | | PMID:31285614 |
| Software, algorithm | RP cone counting | This paper | | MATLAB scripts available at https://github.com/ sawyerxue/RP-cone-count |

Animals *rd1* mice were the albino FVB strain, which carries the *Pde6b*<sup>rd1</sup> allele (MGI: 1856373). BALB/c, CD1, and FVB mice were purchased from Charles River Laboratories. Due to availability, some of the FVB mice were purchased from Taconic, and we did not notice any difference between the two sources in terms of cone degeneration rate. C57BL/6J, *rd10*, and *Parp1*<sup>-/-</sup> mice were purchased from The Jackson Laboratories and bred in house. C57BL/6J and *rd10*, which is on C57BL/6J background, carry a null mutation in the *Nnt* gene (*Freeman et al., 2006*), but not in the other strains (i.e., *rd1*, *Rho*<sup>-/-</sup>, *Parp1*<sup>-/-</sup>, CD1, and BALB/c). We crossed the *Parp1*<sup>-/-</sup> mice with FVB mice to generate homozygous *Parp1*<sup>-/-</sup> *rd1* and *Parp1*<sup>+/+</sup> *rd1* mice. Genotyping of these mice was done by Transnetyx (Cordova, TN). The *Rho*<sup>-/-</sup> mice were kindly provided by Janice Lem (Tufts University, MA) (*Lem et al., 1999*).

## AAV vector design, authentication, and preparation

Detailed information of all AAVs used in this study is listed in *Figure 1—source data 1*, along with the authentication information. cDNAs of mouse Txnip, Hif1a, Hk2, Ldha, Ldhb, Slc2a1, Bsg1, Cpt1a, Oxct1, Mpc1, and Mpc2, and human Nrf2, were purchased from GeneCopoeia (Rockville, MD). Mouse Vegf164 cDNA (*Robinson and Stringer, 2001*) was synthesized through Integrated DNA Technologies (Coralville, IA). We obtained the following plasmids as gifts from various depositors through Addgene (Watertown, MA): Hk1, Pfkm, and Pkm2 (William Hahn and David Root; #23730, #23728, #23757), Pkm1 (Lewis Cantley and Matthew Vander Heiden; #44241), H2BGFP (Geoff Wahl; #11680), mitoRFP (i.e., DsRed2-mito, Michael Davidson; #55838), GFP-Txnip (Clark Distelhorst; #18758), W3SL (Bong-Kiun Kaang; #61463), 3xFLAG (Thorsten Mascher, #55180), and PercevalHR and pHRed (Gary Yellen; #49082, #31473). The cDNA of mouse RdCVF was a gift from Leah Byrne and John Flannery (UC Berkeley, CA). iGlucoSnFR was provided under a Material Transfer Agreement by Jacob Keller and Loren Looger (Janelia Research Campus, VA). RedO promoter was provided as a gift, and SynPVI (also known as ProA7) and SynP136 (also known as ProA1) promoters were provided under a Material Transfer Agreement, from Botond Roska (IOB, Switzerland). The Best1 promoter was synthesized by a lab member, Wenjun Xiong, using Integrated DNA Technologies based on the literature (*Esumi et al., 2009*). Mutated Txnip, dominant-negative HIF1α (*Jiang et al., 1996*), and RO1.7 promoter (*Ye et al., 2016*) were created from the Hif1a and RedO plasmids correspondingly in house using Gibson assembly. Of note, we found that the RedO promoter is stronger than SynP136 or SynPVI promoters, but less specific. RedO has a low level of expression in some rods. SynP136 and SynPVI drive expression that is exclusive to cones, that is, no rod expression in keeping with the observation of Roska Lab (*Jüttner et al., 2019*). SynPVI (0.5 kb) is shorter than SynP136 (2 kb), and is thus better for packaging insert genes that have a large size.

All of the new constructs in this study were cloned using Gibson assembly. For example, AAV-RedO-Txnip was cloned by replacing the EGFP sequence of AAV-RedO-EGFP at the NotI/HindIII sites, with the Txnip sequence, which was PCR-amplified from the cDNA vector adding two 20 bp overlapping sequences at the 5′- and 3′-ends. All of the AAV plasmids were amplified using Stbl3 *Escherichia coli* (Thermo Fisher Scientific). The sequences of all AAV plasmids were verified with

directed sequencing and restriction enzyme digestion. The key plasmids were verified with Next-Generation complete plasmid sequencing (MGH CCIB DNA Core), which is able to capture the full sequence of the ITR regions. The genome sequence of critical AAVs (i.e., AAV8-RedO-Txnip.C247S and AAV8-RedO-Txnip.S308A) was verified with PCR and directed sequencing.

All of the vectors were packaged in recombinant AAV8 capsids using 293 T cells and purified with iodixanol gradient as previously described (*Grieger et al., 2006*; *Xiong et al., 2015*). The titer of each AAV batch was determined using protein gels, comparing virion protein band intensities with a previously established AAV standard. The concentration of our AAV production usually ranged from $2 \times 10^{12}$ to $3 \times 10^{13}$ gc/mL. Multiple batches of key AAV vectors (e.g., four batches of AAV8-RedO-Txnip and three batches of AAV-RedO-siLdhb[(#2)]) were made and tested in vivo to avoid any potential batch effects.

## shRNA

The shRNA plasmids of Ldhb, Slc2a1, Oxct1, and Cpt1a were purchased from GeneCopoeia, provided as three or four distinct sequences for each gene, driven by the H1 or U6 promoter. The knockdown efficiency of these candidate shRNA sequences was tested by co-transfecting with CAG-TargetGene-IRES-d2GFP vector in 293 T cells as previously described (*Matsuda and Cepko, 2007*; *Wang et al., 2014*). The GFP fluorescence intensity served as a fast and direct read out of the knockdown efficiency of these shRNAs. Using this method, we selected the following sense strand sequences to knock down the targeted genes (*Figure 2—figure supplement 2*, *Figure 3—figure supplements 2–4*): siLdhb[(#2)] 5′-CCATCATCGTGGTTTCCAACC-3′; siLdhb[(#1)] 5′-GCAGAGAAATGTCAACGTGTT-3′; siLdhb[(#3)] 5′-GCCGATAAAGATTACTCTGTG-3′; siSlc2a1[(#a)] 5′-GGTTATTGAGGAGTTCTACAA-3′; siOxct1[(#c)] 5′-GGAAACAGTTACTGTTCTCCC-3′; siCpt1a[(#c)] 5′-GCATAAACGCAGAGCATTCCT-3′; and siNC (control sequence that does not target any known gene) 5′-GCTTCGCGCCGTAGTCTTA-3′. We cloned the entire hairpin sequence (including a 6 bp 5′-end lead sequence 5′-gatccg-3′, a 7 bp loop sequence 5′-TCAAGAG-3′ between sense and antisense strands, and a >7 bp 3′-end sequence 5′-ttttttg-3′) and packaged them into AAV8-RedO-shRNA using Gibson assembly as described above. To maximize the knockdown efficacy using a Pol II promoter in AAV (*Giering et al., 2008*), no extra base pairs were retained between the RedO promoter and the 5′-end sequence of shRNAs. Due to the lack of an adequate Ldhb antibody, we confirmed the in vivo Ldhb knockdown efficiency of all three AAV8-RedO-siLdhb vectors by co-injection with an AAV8-Ldhb-3xFLAG vector into WT mouse eyes with detection using FLAG immunofluorescence as described in the Histology section (*Figure 3—figure supplement 1A*).

## Subretinal injection

On the day of birth (P0), AAVs were injected into the eyes of pups as previously described (*Matsuda and Cepko, 2007*; *Xiong et al., 2015*). For all experiments in which cones were quantified, and to provide a means to trace infection (e.g., for IHC), $2.5 \times 10^8$ vg/eye of AAV8-RedO-H2BGFP was co-injected with the experimental AAVs, or alone as a control. The dose of each experimental AAV was $\approx 1 \times 10^9$ vg/eye, individually or when in combination with other experimental AAV's, and the injection volume was the same for each injection. For all other experiments, such as FACS sorting and ex vivo live imaging, $1 \times 10^9$ vg/eye of AAV8-SynP136-H2BGFP, which provides better cone specificity but lower expression level than RedO-H2BGFP, was co-injected. All of the control groups in this study refer to AAV reporter (e.g., H2BGFP or PercevalHR) injection alone.

## Photopic visual acuity measured for optomotor response

The photopic optomotor response of mice was measured using the OptoMotry System (CerebralMechanics) at a background light of $\approx 70$ cd/m$^2$ as previously described (*Xiong et al., 2019*). The contrast of the grates was set to be 100%, and temporal frequency was 1.5 Hz. The threshold of mouse visual acuity (i.e., maximal spatial frequency) was tested by an examiner without knowledge of the control vs. experimental groups. During each test, the direction of movement of the grates (i.e., clockwise or counterclockwise) was randomized, and the spatial frequency of each testing episode was determined by the software. Without knowing the spatial frequency of the moving grates, the examiner reported either 'yes' or 'no' to the system until the threshold of acuity was determined by

the software. Optomotor tests were conducted on *rd10* and *Rho*[-/-] mice, but not with *rd1* strain, which loses vision at a very early age, before any meaningful test could be performed.

## Electroretinography

To probe *rd10* cone function, photopic ERGs were measured in anesthetized mouse eyes in vivo as previously described using an Espion E3 System (Diagonsys LLC) (*Xiong et al., 2019*; *Xue et al., 2020*). Multiple white flashes were applied to elicit ERG responses at 1 (peak), 10 (peak), 100 (xenon), and 1000 (xenon) cd s/m$^2$ intensities with a white light background of 30 cd/m$^2$. A ketamine/xylazine cocktail (100/10 mg/kg) was introduced via intraperitoneal injection to mice for anesthesia. Tropicamide 1% eye solution (Bausch + Lomb) was used to dilate the pupil.

## Histology

Mice were euthanized with $CO_2$ and cervical dislocation, and the eyes were enucleated. For flat-mounts, retinas were separated from the rest of the eye using a dissecting microscope and were fixed in 4% paraformaldehyde solution for 30 min. The retinas were then flat-mounted on a glass slide and coverslip. For H2BGFP-labeled cone imaging, we used a Keyence fluorescence microscope with a ×10 objective (Plan Apo Lamda 10x/0.45 Air DIC N1) and GFP filter box (OP66836).

For cone opsin antibody staining in whole-mount retinas, after fixation, retinas were blocked for 1 hr in PBS with 5% normal donkey serum and 0.3% Triton X-100 at room temperature. After blocking, retinas were incubated with a mixture of 1:200 anti-s-opsin (OPN1MW) antibody (AB5407, EMD Millipore) and 1:600 anti-m-opsin (OPN1SW) antibody (AB5405, EMD Millipore) in the same blocking solution overnight at 4°C, followed by secondary donkey-anti-rabbit antibody staining (1:1000, Alexa Fluor 594) at room temperature for 2 hr, then flat-mounted on a glass slide and coverslip.

For frozen sections, whole eyes were fixed in 4% paraformaldehyde solution for 2 hr at room temperature, followed by removal of the cornea, lens, and iris. The eye cups then went through a 15% and 30% sucrose solution to dehydrate at room temperature, followed by overnight incubation in 1:1 30% sucrose and Tissue-Tek O.C.T. solution at 4°C. Eye cups were embedded in a plastic mold, frozen in a −80°C freezer, and cut into 20 or 12 μm thin radial cross sections that were placed on glass slides. Antibody staining was done similarly to whole-mounts as described above and previously (*Wang et al., 2014*). PBS with 0.1% Triton X-100, 5% normal donkey serum ,and 1% bovine serum albumin (BSA) was used as the blocking solution, except for FLAG detection (10% donkey serum and 3% BSA). GLUT1 (encoded by *Slc2a1* gene) antibody (GT11-A, Alpha Diagnostics) was used at 1:300 dilution, PARP1 antibody (ab227244, Abcam) was used at 1:300 dilution, GFP antibody (ab13970, Abcam) was used at 1:1000 dilution to detect GFP-Txnip, ARR3 antibody (AB15282, Millipore Sigma) was used at 1:1000 dilution to detect cone arrestin (encoded ay the *Arr3* gene), and FLAG antibody (ab1257, Abcam) was used at 1:2000 based on a previous study (*Ferrando et al., 2015*). If applicable, 1:1000 PNA (CY5 or rhodamine labeled) for cone extracellular matrix labeling and 1:1000 DAPI were used to co-stain with secondary antibodies. Stained sections were imaged with a confocal microscope (LSM710, Zeiss) using ×20 or ×63 objectives (Plan Apo 20x/0.8 Air DIC II, or Plan Apo 63X/1.4 Oil DIC III).

## Automated cone counting

The cone-H2BGFP images of entire flat-mounted retinas were first analyzed in ImageJ to acquire the diameter and the center parameters of the sample. We used a custom MATLAB script to automatically count the number of H2BGFP-positive cones in the central ½ radius of the retina since RP cones degenerate faster in the central than the peripheral retina. The algorithm was based on a Gaussian model to identify the centers of labeled cells (*Wu et al., 2021*). The threshold of peak intensity and the variance of distribution were initially determined using visual inspection, and a comparison to the number of manually counted cones from six retinas. The threshold of intensity and variance thus determined were then set at fixed values for all the experiments that used cone quantification. The background intensity did not interfere with the accurate counting on the raw images by this MATLAB script despite the observation that the intensity of some images looked different at low magnification. This method of cone counting was further verified using FACS analysis of cones in this study (*Figure 1—figure supplement 2B*) and an independent study (*Wang et al., 2020*).

## Live imaging of cones on ex vivo retinal explants

For JC-1 mitochondrial dye staining, the retina was quickly dissected in a solution of 50% Ham's F-12 Nutrient Mix (11765054, Thermo Fisher Scientific) and 50% Dulbecco's Modified Eagle Medium (DMEM; 11995065, Thermo Fisher Scientific) at room temperature. They were then incubated in a culture medium containing 50% Fluorobrite DMEM (A1896701, Thermo Fisher Scientific), 25% heat-inactivated horse serum (26050088, Thermo Fisher Scientific), and 25% Hanks' Balanced Salt Solution (14065056, Thermo Fisher Scientific) with 2 µM JC-1 dye (M34152, Thermo Fisher Scientific) at 37°C in a 5% $CO_2$ incubator for 20 min. The retinas were washed in 37°C in this medium without JC-1 three times, transferred to a glass-bottom culture dish (MatTek P50G-1.5-30F) with culture medium, and imaged using a confocal microscope (LSM710 Zeiss), which was equipped with a chamber pre-heated to 37°C with pre-filled 5% $CO_2$. Right before imaging, a cover slip (VWR 89015-725) was gently applied to flatten the retina. Regions of interest (with H2BGFP as an indicator of successful AAV infection and to set the correct focal plane on the cone layer) were selected under the eyepiece with a ×63 objective (Plan Apo 63X/1.4 Oil DIC III). Fluorescent images from the same region of interest were obtained with the excitation wavelength of 561 nm (for J-aggregates), 514 nm (for JC-1 monomer), and 488 nm (for H2BGFP). Four different regions of interest from the central part of the same retina were imaged before moving to the next retina.

For RH421 ($Na^+/K^+$ ATPase dye) staining, similar steps were taken as for JC-1 staining, with the following modifications: (1) 0.83 µM RH421 dye (61017, Biotium) was added to the glass-bottom culture dishes just before imaging, but not during incubation in the incubator, due to the fast action of RH421. (2) Five regions of interest were imaged per retina from the central area. (3) The dissection and culture medium were lactate-only medium (see below). (4) Excitation wavelengths: 561 nm (RH421) and 488 nm (H2BGFP).

For imaging genetically encoded metabolic sensors (PercevalHR, iGlucoSnFR, and pHRed), retinas were placed in the incubator for 12 min and then taken to confocal imaging without any staining. For the high-glucose condition, the culture medium described above contains ≈15 mM glucose without lactate or pyruvate. For the lactate-only condition, the culture and dissection media were both glucose-pyruvate-free DMEM (A144300, Thermo Fisher Scientific) and were supplemented with 20 mM sodium L-lactate (71718, Sigma-Aldrich). For the pyruvate-only condition, the culture and dissection media were both glucose-pyruvate-free DMEM plus 10 or 20 mM sodium pyruvate (P2256, Sigma-Aldrich). No AAV-H2BGFP was co-injected with these sensors since the sensors themselves could be used to trace the area of infection. The excitation wavelengths for sensors were 488 and 405 nm (PercevalHR, ratiometric high and low ATP:ADP), 488 and 561 nm (iGlucoSnFR, glucose-sensing GFP and normalization mRuby), and 561 and 458 nm (pHRed, ratiometric low and high pH).

The fluorescent intensity of all acquired images was measured by ImageJ. The ratio of sensors/dye was normalized to averaged control results taken at the same condition.

## Flow cytometry and cell sorting

All flow cytometry and cell sorting were performed on MoFlo Astrios EQ equipment. Retinas were freshly dissected and dissociated using cysteine-activated papain followed by gentle pipetting (*Shekhar et al., 2016*). Before sorting, all samples were passed through a 35 µm filter with buffer containing Fluorobrite DMEM (A1896701, Thermo Fisher Scientific) and 0.4% BSA. Cones labeled with AAV8-SynP136-H2BGFP (highly cone-specific) were sorted into the appropriate buffer for either ddPCR or RNA sequencing.

## RNA sequencing

RNA sequencing was done as previously described (*Wang et al., 2019*). 1000 H2BGFP-positive cones per retina were sorted into 10 µL of Buffer TCL (QIAGEN) containing 1% β-mercaptoethanol and immediately frozen in −80°C. On the day of sample submission, the frozen cone lysates were thawed on ice and loaded into a 96-well plate for cDNA library synthesis and sequencing. A modified Smart-Seq2 protocol was performed on samples by the Broad Institute Genomics Platform with ~6 million reads per sample (*Picelli et al., 2013*). The reads were mapped to the GRCm38.p6 reference genome after quality control measures. Reads assigned to each gene were quantified using featureCounts (*Liao et al., 2014*). Count data were analyzed using DESeq2 to identify differentially expressed genes, with an adjusted p value less than 0.05 considered significant (*Anders and Huber,*

*2010*). The raw results have been deposited to Gene Expression Omnibus (accession number GSE161622 for RP cones and GSE168503 for WT cones).

## ddPCR

RNA was isolated from 20,000 sorted cones per retina using the RNeasy Micro Kit (QIAGEN) as previously described (*Wang et al., 2020*) and converted to cDNA using the SuperScript III First-Strand Synthesis System (Invitrogen). cDNA from each sample was packaged in droplets for Droplet Digital PCR (ddPCR) using QX200 EvaGreen Supermix (#1864034). The reads of expression were normalized to the housekeeping gene *Hprt*. Sequences for RT-PCR primers were designed using the IDT online RealTime qPCR primer design tool. The following primers were selected for the genes of interest: *Txnip* (forward 5′-ACATTATCTCAGGGACTTGCG-3′; reverse 5′-AAGGATGACTTTC TTGGAGCC-3′), *Hprt* (forward 5′-TCAGTCAACGGGGGACATAAA-3′; reverse 5′-GGGGCTGTAC TGCTTAACCAG-3′), *mt-Nd4* (forward 5′-AGCTCAATCTGCTTACGCCA-3′; reverse 5′-TG TGAGGCCATGTGCGATTA-3′), *mt-Cytb* (forward 5′-ATTCTACGCTCAATCCCCAAT-3′; reverse 5′-TATGAGATGGAGGCTAGTGGC-3′), *mt-Co1* (forward 5′-TCTGTTCTGATTCTTTGGGCACC-3′; reverse 5′-CTACTGTGAATATGTGGTGGGCT-3′), *Acsl3* (forward 5′- AACCACGTATCTTCAACACCA TC-3′; reverse 5′- AGTCCGGTTTGGAACTGACAG-3′), and *Ftl1* (forward 5′- CCATCTGACCAACC TCCGC-3′; reverse 5′- CGCTCAAAGAGATACTCGCC-3′).

## Transmission electron microscopy

Intracardial perfusion (4% PFA + 1% glutaraldehyde) was performed on ketamine/xylazine (100/10 mg/kg) anesthetized mice before the removal of eyes. The cornea was sliced open and the eye was fixed with a fixative buffer (1.25% formaldehyde + 2.5% glutaraldehyde + 0.03% picric acid in 0.1 M sodium cacodylate buffer, pH 7.4) overnight at 4℃. The cornea, lens, and retina were removed before resin embedding, ultrathin sectioning, and negative staining at Harvard Medical School Electron Microscopy Core. The detailed methods can be found on the core's website (https://electron-microscopy.hms.harvard.edu/methods). The stained thin sections were imaged on a conventional TEM (JEOL 1200EX) with an AMT 2k CCD camera.

## Statistics

For the comparison of two sample groups, two-tailed unpaired Student's t-test was used to test for the significance of difference, except for P140 $Rho^{-/-}$ optomotor assay (paired two-tail t-test). For comparison of more than two sample groups, ANOVA and Dunnett's multiple comparison test were performed in Prism 8 software to determine the significance. A p-value of <0.05 was considered statistically significant. All error bars are presented as mean ± standard deviation, except for the *rd10* optomotor assays, ERG, FACS cell %, and RNA-seq raw reads (mean ± SEM).

## Study approval

All animal experiments were approved by the IACUC of Harvard University in accordance with institutional guidelines.

## Acknowledgements

We thank Sui Wang, ChangHee Lee, Sylvain Lapan, Gabby Niconchuk, Brian Rabe, Cem Sengel, Sophia Zhao, Yuji Atsuta, Wenjun Xiong, Ryoji Amamoto, Grace Wallick, Gary Yellen, Zhongjie Fu, Zhengping Hu, Maryna Ivanchenko, Paula Montero-Llopis, Microscopy Resources on the North Quad, Maria Ericsson, Electron Microscopy Facility, Flow Cytometry of Immunology, Marcelo Cicconet, Image and Data Analysis Core of Harvard Medical School, Genomics Platform of Broad Institute, Metabolomics Core Resource Laboratory of New York University, and Frans Vinberg at University of Utah for discussions and technical support. We also thank Botond Roska, Jacob Keller, Loren Looger, Leah Byrne, John Flannery, William Hahn, David Root, Lewis Cantley, Matthew Vander Heiden, Geoff Wahl, Michael Davidson, Clark Distelhorst, Bong-Kiun Kaang, and Thorsten Mascher for plasmids. This work was funded by the National Institute of Health (K99EY030951 to YX and U01EY025497 to CLC), Alcon Research Institute (CLC), Astellas Pharmaceuticals (CLC), and Howard Hughes Medical Institute (CLC).

## Additional information

### Funding

| Funder | Grant reference number | Author |
|---|---|---|
| National Eye Institute | K99EY030951 | Yunlu Xue |
| National Eye Institute | U01EY025497 | Constance L Cepko |
| Alcon Research Institute | | Constance L Cepko |
| Astellas Pharmaceuticals | | Constance L Cepko |
| Howard Hughes Medical Institute | | Constance L Cepko |

The funders had no role in study design, data collection and interpretation, or the decision to submit the work for publication.

### Author contributions

Yunlu Xue, Conceptualization, Resources, Data curation, Software, Formal analysis, Funding acquisition, Validation, Investigation, Visualization, Methodology, Writing - original draft, Writing - review and editing; Sean K Wang, Resources, Formal analysis, Validation, Investigation, Writing - review and editing; Parimal Rana, Formal analysis, Validation, Investigation; Emma R West, Software, Formal analysis; Christin M Hong, Formal analysis, Investigation, Methodology, Writing - review and editing; Helian Feng, Formal analysis; David M Wu, Resources, Software, Writing - review and editing; Constance L Cepko, Conceptualization, Resources, Supervision, Funding acquisition, Writing - original draft, Writing - review and editing

### Author ORCIDs

Yunlu Xue (ID) https://orcid.org/0000-0002-2088-9826
Constance L Cepko (ID) https://orcid.org/0000-0002-9945-6387

### Ethics

Animal experimentation: All animal experiments were approved by the IACUC of Harvard University in accordance with institutional guidelines (protocol number: IS1695).

### Decision letter and Author response

Decision letter https://doi.org/10.7554/eLife.66240.sa1
Author response https://doi.org/10.7554/eLife.66240.sa2

## Additional files

### Supplementary files

• Transparent reporting form

### Data availability

Sequencing data have been deposited in GEO under accession codes GSE161622 and GSE168503.

The following datasets were generated:

| Author(s) | Year | Dataset title | Dataset URL | Database and Identifier |
|---|---|---|---|---|
| Xue Y, Cepko CL | 2020 | RP mouse cone transcriptom change with Txnip treatment | https://www.ncbi.nlm.nih.gov/geo/query/acc.cgi?acc=GSE161622 | NCBI Gene Expression Omnibus, GSE161622 |
| Xue Y, Cepko CL | 2021 | Wildtype (wt) mouse cone transcriptome change with Txnip treatment | https://www.ncbi.nlm.nih.gov/geo/query/acc.cgi?acc=GSE168503 | NCBI Gene Expression Omnibus, GSE168503 |

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
