## [Decision Letter]

**Acceptance summary:**

Your screen of several candidate genes that might make cones more robust to stress caused by genetic deficiencies led to the identification of Thioredoxin-interacting protein (Txnip) that is very effective at prolonging cone survival in models of retinal degeneration. Txnip appears to be switching the fuel source and enhancing lactate catabolism in cone photoreceptors, thus allowing cones to use alternative fuels more effectively. This study has the potential to become a therapeutic approach to treat retinal degeneration

**Decision letter after peer review:**

Thank you for submitting your article "AAV-Txnip prolongs cone survival and vision in mouse models of retinitis pigmentosa" for consideration by *eLife*. Your article has been reviewed by 3 peer reviewers, and the evaluation has been overseen by a Reviewing Editor and Marianne Bronner as the Senior Editor. The following individual involved in review of your submission has agreed to reveal their identity: James B Hurley (Reviewer #1).

The reviewers, who have different expertise, have discussed their reviews with one another, and the Reviewing Editor has drafted this to help you prepare a revised submission.

The three reviewers agree that the study is significant and might lead to potential clinical applications. However, they also noted some weaknesses that must be addressed. These include missing controls and quantifications that would reinforce the data.

I am sure that you will do your best to address many of the comments from these very thorough reviews but there is a small number of points that must absolutely be addressed:

– You should demonstrate convincingly that the proteins, especially Txnip and GLUT1 are expressed and that the levels of their over-expression compared to endogenous expression are substantial.

– Although you do show a rescue of cone survival, you must address whether the structure and function of these cones are indeed maintained. You should present data about the shape of the outer segments and about the cone synapses (in P50 retina?). Histology, IHC or TEM if feasible.

– Is TXNIP's rescue mechanistically mediated through anti-oxidative stress or mitochondrial biogenesis? Checking the difference in the number of mitochondria in Txnip-treated retina would be helpful.

Reviewer #1: (Recommendations for the authors)

1. Injections of the Txnip gene from AAV with cone-specific promoters slows retinal degeneration. A control that is missing throughout this very interesting and significant report is direct evidence that Txnip actually is being overexpressed. My confidence in the conclusions would be stronger if the authors could show by immunoblot or by IHC that there is normally some endogenous expression of Txnip in cones and that when the eyes have in them AAV carrying a cone-specific promoter and the coding sequence for Txnip that Txnip expression in cones increases substantially. It would be very helpful if the level of over-expression could be quantified.

2. In theory when Txnip is over-expressed in cones there should be a decrease in GLUT1 expression in cones, but not in RPE cells. This report would be stronger if the authors could show by IHC that cone expression of GLIT1 is diminished in the Txnip cone OE retinas but GLUT1 would not be diminshed in the cones that express the mutant Txnip that is incapable of stimluating endocytosis of GLUT1.

3. Lines 259-261. The finding that expression of ldhb prolongs survival of cones in these and other experiments throughout the report is interesting. It implies that cones are using lactate. But I would expect when rods are gone there is not much lactate in the retina. Please check that – i.e. measure lactate levels in the WT and rd1 retinas. The assay could be done with a commercial kit. If the lactate levels are low in the rd1 retina then it is puzzling why enhanced ldhb expression prolongs survival. Please address this either here or in the discussion. Also, it would be helpful to confirm with an antibody that ldhb is being expressed (in the treated retinas at higher than normal levels).

4. I think readers would like to know what happens to the phosphorylation state of mTOR in cones in these experiments and I think it would be relatively simple to test. In some cells phosphorylation of mTOR in the absence of glucose can be harmful, I think because it is prepping cells for a metabolic state that does not exist in the tissue. The Cepko lab reported previously that mTOR phosphorylation in cones is diminished in rd mutants, consistent with a compromised supply of glucose to the retina. What effect does Txnip expression in cones have on mTOR phosphorylation (diminish, increase or no effect) and what effect does combined over-expression of HK1 and PFK1 expression have on mTOR phosphorylation?

5. Lines 277-286. Please confirm by IHC or immunoblot that the wt or dnHIF1 is over-expressed and, if possible, quantify by how much it is over-expressed.

6. Extended data 6a and lines 300-302. I would have expected GLUT1 to be on the basal and apical surfaces of the Txnip.CS.LLAA mutant overexpressor but it seems to be only on the apical surface. Please mention and discuss this.

Reviewer #2 (Recommendations for the authors):

The manuscript includes extensive data. However, several points require clarification and the text and figures need consistency. Specific comments/suggestions are as follows:

The authors concluded previously that cone starvation and degeneration are due to shortage of glucose in four animal models of RP (Punzo et al., 2009). However, except rho-/-, which is not the focus in this manuscript, the rest three animal models were not used in this study. It is unclear whether rd1 and rd10 also have shortage of ATP and how severe the shortage is as compared to the WT, or it is simply more ATP can better maintain cone survival.

The authors claim the treatment "correlates with the presence of a healthier mitochondria". As shown in Figure 5a, although the size of mitochondria was bigger in the treated groups compared to the control, those mitochondria did not seem to harbor a typical morphology. More transmission electron microscopy (TEM) images of different magnitudes are needed to show the localization of these mitochondria as well as the morphology of more mitochondria. Without the images from WT, it is hard to evaluate whether it is due to strain of the mice or other factors, or how a healthy mitochondrion should look like. Likewise, a much higher membrane potential in mitochondria does not necessarily mean healthier mitochondria. Additionally, the authors should show the mitochondria morphology of P50 rd1 as well, as that's the end point of the whole experiments and can provide useful information on the duration of the treatment.

Although the flat mount images in this manuscript show the overall cell survival in the whole retina, immunostaining of retinal sections is needed to address important questions related to retinal functions such as the structure of the retina, the morphology of the survival cone photoreceptors and the subtypes of the surviving cones, i.e., whether Txnip can maintain both S- and M- cones. The authors mentioned "this combination also preserved the RP cone outer segments" (line 312); yet, it is hard to evaluate the outer segment by flat mount, although a slight difference may be noticeable between the two groups.

The automated method for cone quantification (1/2 radius cell count) that counts H2BGFP-positive cones in the central half of the retina (previous calculation of the diameter of flat mount retinas) raises some concerns. Is the area of the retina in the 1/2 radius the same between samples? Not all flat-mounted retinas look the same (differences due to retina dissection), and this can influence the area used to count cones. In addition, is this method counting all the cones present in the sample? Are all cones H2BGFP-positive? What is the viral transduction efficiency (alone and co-injecting two AAVs)? Is it the same between samples? Are authors using z-stack confocal images for quantification?

It is appropriate to quantify density of cones (number of cones/mm2) in defined areas of interest, which should be the same for all the samples. To achieve this, a proper orientation of the whole-mounted retina is necessary, and this information is lacking in the images included in the manuscript.

The authors could consider the quantification of "holes" or "craters" in the cone mosaic as an indicator of differential treatment effect.

The authors do not quantify the level of Txnip expression in transduced retinas. It would be important to know the level of overexpression they have achieved and compare it with the expression in untreated RP retinas. In addition, it would be relevant to know if treatment effects correlate with Txnip expression levels.

OMR (based on head movements of mice observed by a human observer) can't be directly equated to visual acuity (see PLoS ONE 8(11): e78058, 2013; J Neurophysiol 118: 300-316, 2017). One needs to interpret such data with caution. Is it possible to do multi-focal ERG?

The authors confirmed the effect of Txnip on cone survival using three different animal models. However, in some follow-up experiments, the authors used these models interchangeably and it leads to confusion. For example, while rd1 is the main focus of the manuscript, the authors did not show the rescued visual acuity of these mice. Even if there is no effect of Txnip on visual acuity in these mice, the authors should still mention this to provide some information of the effectiveness of the treatment on some fast-degenerating mouse models. In addition, the authors used rd10 and rho-/- as well in the manuscript. Did the authors test the mechanisms they found using rd1 on rd10 and rho-/-? These data would be helpful to discriminate whether the switch of energy source by Txnip in the survival of cone cells is specific to rd1.

Figure 8a and 8b described the effect of vector combinations on cone survival in rd1, whereas Figure 8c actually mentioned the effect on outer segment in rd10. The authors did not even mention this switch of animal models in the text. The effect on outer segment in rd10 mice does not mean it would surely give the same effect in rd1 mice. It is unclear whether there was no improvement of outer segment in treated rd1 mice, or the vector combination did not have any effect on cone survival in rd10 mice.

Reviewer #3 (Recommendations for the authors):

In order to strengthen this manuscript, the following suggestions should be addressed:

1. Did the authors observe the difference in mice with or without NNT deletion? Is there an NNT deletion in rd1, rd10, parp1-/-, and rho-/- mice?

2. Extended Data Figure 1b and 1d: to show AAV8-RO1.7-GFP expression in cone photoreceptors, the Arr3 antibody will be better than PNA since PNA could only express in the cone outer and inner segments. Extended Data Figure 1b showed some GFP in the cytoplasm, is this the H2B-EGFP?

3. Figure 1b: are these numbers within the 1/2 radius or within selected small boxes? If these numbers are from selected small boxes, how many small boxes from each were selected? The authors should explain how they selected the small boxes as there is some variation on the dorsal retina or ventral retina. For example, in P130 rd10, the cone cell densities are quite different in the four leaves of the inner circle. Same question for the P50 rd1, there is some area with a patchy absence of cone cells. If the numbers are within 1/2 radius, why use 1/2 radius rather than 2/3 or 1/3 radius?

4. It would be helpful if the authors explained the reason behind each time point, i.e., P50 for rd1, P130 for rd10, and P150 for Rho -/-.

5. Figure 5i: the number of cone cells at 1/2 radius in the whole mount retina looks quite similar, but it seems the number of cone cells in the periphery of control retina is more than the number in the +Ldhb retina.

6. Bracket "[]" in Figure 4, and curly bracket "{}" in Figure 5 indicate sample size. Are these the sample sizes of the image/retina or samples sizes of the mice?

7. Figure 5: it is unclear how reliable the Mt size is, because the EM images are not SBF-SEM. The shape and morphology of mitochondria is in 3D and the EM images are taken from a different orientation and thus may have different results.

8. Did the authors observe any difference in the number of mitochondria in Txnip-treated retina?

9. Does Txnip increase mitochondrial biogenesis?

10. It would be better if the authors could clarify the meaning of n.d. in "Wu et al., n.d.".

11. Line 336-337: it remains unclear what the role of anti-oxidative stress is in the Txnip-mediated cone rescue effect. It would be helpful if the authors added an experiment to demonstrate the changes in ROS in Txnip (wt and C237S)-mediated activities.

12. Line 353: it would be helpful if the authors could explain why the Txnip rescue did not depend on ketolysis or ß-oxidation.

13. Line 487-488: "The GFP fluorescence intensity served as a fast 488 and direct read out of the knockdown efficiency of these shRNAs." The GFP after IRES can sometimes too weak to produce a fast and direct readout. Do the authors have the image/a figure on this?

14. It is unclear whether the control group in this study refers to the non-injected eye or to saline injections.

---

## [Author Response]

The three reviewers agree that the study is significant and might lead to potential clinical applications. However, they also noted some weaknesses that must be addressed. These include missing controls and quantifications that would reinforce the data.I am sure that you will do your best to address many of the comments from these very thorough reviews but there is a small number of points that must absolutely be addressed:– You should demonstrate convincingly that the proteins, especially Txnip and GLUT1 are expressed and that the levels of their over-expression compared to endogenous expression are substantial.

Since our gene therapy is highly specific to a small population of cells in a heterogenous tissue (i.e. cones in the degenerating retina), the only protein detection method that might work would be IHC. We had repeatedly tried IHC for Txnip with two commercially available antibodies at the early stage of this study. The best images are shown in Author response image 1, but are not convincing. The best of the available Txnip antibodies (Thermo Fisher #40-3700) was able to detect Best1-Txnip overexpression in the RPE, where there is low staining in the uninfected tissue and high vector expression from the Best1 promoter (Author response image 1). However, this antibody gave ambiguous staining in the retina (Author response image 1). There is almost no Txnip RNA in cones (new Figure 5—source data 3), almost none in rods (Drop-seq data from Shekhar et al., 2017), and a very low signal in a small fraction of Mueller glia (Dropseq data from Shekhar et al., 2017). Yet, quite a bit of IHC signal was detected in the outer nuclear layer and inner and outer segment layers in the uninfected retina, with no obvious greater IHC signal in the AAV-Txnip infected retina. Due to these problems, we made a fusion protein of GFP-Txnip and used AAV8-RedO1.7 to express it, in both RP and WT retinas (Figure 1—figure supplement 1B, originally it was shown in Extended Data Figure 1b). We found that GFP-Txnip was excluded from cone outer segments, but was located in all other regions, including in the reported locations of a protein with which it is reported to interact (PARP1), in the inner segments and nucleus.

**Author response image 1. sa2fig1:** 

For GLUT1 IHC, the antibody was not the issue. We had shown significant removal of GLUT1 from the RPE by Best1-Txnip in the original submission (Figure 7—figure supplement 1, originally Extended Data Figure 6a; Author response image 2). However, since GLUT1 is a membrane protein and cones are surrounded by the processes of neighboring cells (i.e. rods, Muller glia, ad RPE in WT retina; Muller glia and RPE processes in the RP retina after rods are gone), and because all of these other cell types express quite a bit of GLUT1, we had trouble distinguishing GLUT1 IHC signals within cones (Author response image 2 at the end of this letter). For this reason, we could not tell if there was any drop in cone GLUT1 IHC by RedO-Txnip (Author response image 2). Moreover, the raw reads from our RNA-seq dataset suggested that the GLUT1 (i.e. *Slc2a1*) level was relatively low in cones (new Figure 5—source data 4). Therefore, we did not think that we would be able to observe a significant GLUT1 reduction in Txnip-treated cones, as was observed in the RPE.

We did not include these IHC results given that they were inconclusive. In addition to the RNA reads for GLUT1, we have included a table of Txnip raw reads from our cone RNAseq data (GEO depositions: GSE161622 and GSE168503) of Txnip-treated vs. control of four different strains (new Figure 5—source data 3). These numbers clearly demonstrate >100 fold increase of Txnip mRNA through addition of RedO-Txnip in all strains. The Txnip ddPCR data confirmed this finding (Figure 5—figure supplement 1B, originally Extended Data Figure 5b). As mentioned above, there is almost no Txnip mRNA in wildtype or RP control cones, so the fold amplification is huge. Although these data are for RNA, and not protein, (and we would have loved to be able to show protein), the only alternative hypothesis to protein expression leading to rescue is that the AAV DNA, encoding Txnip, or the mRNA for Txnip, caused the rescue. These are highly unlikely scenarios given that we made several alleles of Txnip that had only base pair changes (e.g. C247S, S308A). These base pair changes affected the rescue activity. The most likely hypothesis is that these base pair changes resulted in protein changes, as otherwise, we would have to conclude that base pair changes to the AAV DNA or mRNA could both increase (C247S) and decrease rescue (S308A). These base pair changes were chosen as the literature had shown that the corresponding amino acid changes changed the activity and binding partners for the Txnip protein.In addition to the efforts described above, we validated the DNA sequences of our AAV vectors. (Figure 1—source data 1, originally Supplementary Table 1, and Methods) using several methods.

– Although you do show a rescue of cone survival, you must address whether the structure and function of these cones are indeed maintained. You should present data about the shape of the outer segments and about the cone synapses (in P50 retina?). Histology, IHC or TEM if feasible

We did not observe any obvious structural changes to either RP or WT cones by Txnip, and we have not claimed any RP cone structural benefits in any part of our text. In fact, we have never seen nice outer segments or wildtype cone-like morphology for any of our rescue treatments, nor from those of other laboratories. We have seen more opsin protein, and a bit of an outer segment structure with Txnip C247S (Figure 6C). Our best guess is that, given the complete collapse of the outer nuclear layer (ONL), the cones are challenged by mechanical/physical changes in the architecture of the ONL/RPE interface. In addition, there are data from our lab and others that there are many problems for cones in the RP environment, including oxidative stress, inflammation, and metabolic challenges. Our lab has now assayed >50 genes to address these challenges, and we have been publishing some of the successes (Punzo et al. 2009, Xiong et al. 2015, Wang et al. 2019, 2020, Chinchore et al. 2019, Wu et al. 2021). Our goal is to combine treatments to try to combat multiple problems for cones. Consistent with this hypothesis, we did achieve greater rescue by combining RedOTxnip (for cone metabolism) with Best1-Nrf2 (to combat oxidative damage and possibly inflammation in the RPE) (Figure 8A,B). We believe that the most important way to assess a benefit to cones is to show functional benefits, which is the point of the optomotor assay (Figure 1C), where we did see significant benefit due to Txnip.

However, to address the question of cone morphology, we have added a set of cone arrestin (ARR3) IHC and TEM images to show what we observed (Figure 1—figure supplement 2A, Figure 5—figure supplement 2), as well as IHC for s- and m- opsin separately (Figure 8—figure supplement 1A), and updated the text to address this point. For cone function, we have added our ERG data as Figure 1—figure supplement 2E. The interpretation of this neutral result is in the corresponding point to Reviewer #2 in this letter below.

– Is TXNIP's rescue mechanistically mediated through anti-oxidative stress or mitochondrial biogenesis? Checking the difference in the number of mitochondria in Txnip-treated retina would be helpful.

Mitochondria are thought to be one continuous structure, although when seen in cross section they appear to be independent units. It is thus hard to quantify their numbers. The staining with mitoRFP is the best proxy we could come up with to quantify the Txnip effect on “numbers” of mitochondria, as we reported in the initial submission (Figure 5—figure supplement 1D, originally Extended Data Figure 5d). It is also obvious from cross sections that the Txnip-treated cells have larger mitochondrial cross sections, as quantified in Figure 5B. The TEM images that we showed in the first submission, along with the quantification of the average size of the cross sections, show that Txnip increases the size. We added 22 more TEM images to share our observations on the mitochondrial morphology in the cones from both WT and RP retinas (new Figure 5—figure supplement 2B-D). We found it difficult to compare mitochondria in wildtype healthy inner segments/ OPL to those in degenerating cones due to the collapse of the normal cone morphology i.e. there are no organized inner segments in RP cones, which is where most mitochondria localize in wild type cones.

We observed that Txnip did not upregulate the mitochondrial ETC genes in WT cones as it did in RP cones. Txnip is thus unlikely to have as its constitutive role mitochondria biogenesis. We have added this information to the revised manuscript (new Figure 5—figure supplement 2A, and Figure 5—source data 2). This additional WT cone RNA-seq results have been recently deposited to GEO (accession number GSE168503). However, it is hard to know whether the benefits to RP cone mitochondria are due to increased biogenesis or other effects. Cone degeneration in the *rd10* strain has been reported to be accompanied by swollen and abnormal mitochondria, which are rescued by a RipK3 antagonist (Murakami et al., 2012, PNAS). As this drug can inhibit necrotic death due to oxidative stress, which is mediated by RipK3, we understand that this drug, as well as Txnip, might protect cones via inhibition of oxidative damage or death induced by oxidative stress. Of interest regarding this possibility is that the Txnip C247S allele does not bind thioredoxin, and it rescues better. It could be that by liberating thioredoxin, it allows thioredoxin to increase its anti-oxidation role. Alternatively, or additionally, it could be that Txnip is liberated from thioredoxin to better perform its other role(s) for cone rescue.

Regarding the role of Txnip in oxidative damage, it has been reported to inhibit the thioredoxins when it binds to them, making oxidative damage worse. This would predict that reduction or removal of Txnip would reduce oxidative damage, i.e. an effect in the opposite direction to the benefits that we observe. In keeping with this, Txnip KO Muller glial cells in culture have less mitophagy, in a condition used to model diabetes by culturing in a high glucose medium. Similarly, injection of Txnip shRNAs intravitreally leads to less mitophagy in Muller glia in a diabetic model. More work clearly needs to be done to investigate how Txnip might provide benefits to cone mitochondria, including investigation of the role of thioredoxins, Txnip, and mitochondrial biogenesis or mitophagy.

Reviewer #1: (Recommendations for the authors)1. Injections of the Txnip gene from AAV with cone-specific promoters slows retinal degeneration. A control that is missing throughout this very interesting and significant report is direct evidence that Txnip actually is being overexpressed. My confidence in the conclusions would be stronger if the authors could show by immunoblot or by IHC that there is normally some endogenous expression of Txnip in cones and that when the eyes have in them AAV carrying a cone-specific promoter and the coding sequence for Txnip that Txnip expression in cones increases substantially. It would be very helpful if the level of over-expression could be quantified.

Please refer to our response to the Editor (point #1) above. The relevant data and additional data can be found in Author response image 1 for Txnip IHC at the end of this letter, a new Figure 5—source data 3 for RNA-seq raw reads, and Figure 5—figure supplement 1B (originally Extended Data Figure 5b) for ddPCR.

2. In theory when Txnip is over-expressed in cones there should be a decrease in GLUT1 expression in cones, but not in RPE cells. This report would be stronger if the authors could show by IHC that cone expression of GLIT1 is diminished in the Txnip cone OE retinas but GLUT1 would not be diminshed in the cones that express the mutant Txnip that is incapable of stimluating endocytosis of GLUT1.

Please refer to our response to Editor (point #1) above. The relevant data and additional data can be found in Author response image 2 for GLUT1 IHC at the end of this letter, and a new Figure 5—source data 4 for RNA-seq raw reads.

3. Lines 259-261. The finding that expression of ldhb prolongs survival of cones in these and other experiments throughout the report is interesting. It implies that cones are using lactate. But I would expect when rods are gone there is not much lactate in the retina. Please check that – i.e. measure lactate levels in the WT and rd1 retinas. The assay could be done with a commercial kit. If the lactate levels are low in the rd1 retina then it is puzzling why enhanced ldhb expression prolongs survival. Please address this either here or in the discussion. (Also, it would be helpful to confirm with an antibody that ldhb is being expressed (in the treated retinas at higher than normal levels).

Our data suggest that Txnip reprograms the cones via an unknown mechanism to use lactate more effectively than usual. This does not necessitate more Ldhb protein, but we too wondered if there might be more Ldhb protein in cones following Txnip overexpression. We tried LDHB antibodies from Sigma-Aldrich (SAB2108609) and Thermo Fisher Scientific (PA543141) for IHC, but neither worked well (i.e. low signal-to-noise ratio). This is the reason we created AAV8-RedO1.7-Ldhb-FLAG, and IHC stained against FLAG to test the effectiveness of the Ldhb shRNAs (Figure 3—figure supplement 1A, originally Extended Data Figure 3a). As we are not aware of any function of Ldhb beyond the conversion of lactate to pyruvate, our interpretation of the shRNA knockdown data is that lactate conversion to pyruvate via Ldhb, which is required for Txnip rescue. In support of this, overexpression of Ldha reduced the rescue by Txnip, which is consistent with the conversion of lactate to pyruvate as providing a benefit to cones (Figure 3—figure supplement 1D,E, originally Extended Data Figure 3d,e).Measurement of lactate levels in the retina would reveal the lactate level of the entire retina. As cones are a small fraction of total cells, and we do not know how much lactate is required or where, we do not think that measurements of total retinal lactate would be interpretable. Interestingly, the lactate level is lower than the glucose level in mouse serum, yet the turnover rate of lactate is much higher than it is for glucose in circulation, to feed the TCA cycle of non-CNS organs e.g. muscle (Hui S, 2017, Nature). As our manipulation is specific to a small percentage of cells, it would be challenging to measure the lactate consumption in Txnip treated cones.

4. I think readers would like to know what happens to the phosphorylation state of mTOR in cones in these experiments and I think it would be relatively simple to test. In some cells phosphorylation of mTOR in the absence of glucose can be harmful, I think because it is prepping cells for a metabolic state that does not exist in the tissue. The Cepko lab reported previously that mTOR phosphorylation in cones is diminished in rd mutants, consistent with a compromised supply of glucose to the retina. What effect does Txnip expression in cones have on mTOR phosphorylation (diminish, increase or no effect) and what effect does combined over-expression of HK1 and PFK1 expression have on mTOR phosphorylation?

We would also like to know, but we have not been able to find an antibody that shows the mTOR phosphorylation state since our original publication of this observation. The antibody that we used in that study no longer exists, and though we and Claudio Punzo (who originally did that work) have tried to find another one that works, we have been unable to do so.

5. Lines 277-286. Please confirm by IHC or immunoblot that the wt or dnHIF1 is over-expressed and, if possible, quantify by how much it is over-expressed.

We tried one HIF1A antibody from Novus Biologicals (NB100-479), and found a high background signal for IHC in the RP retina. In our previous study (Punzo et al., 2009), we found that RP cones showed stabilized HIF1A. Even if we had a suitable antibody, this signal might mask overexpression of Hif1a or dnHIF1A.

6. Extended data 6a and lines 300-302. I would have expected GLUT1 to be on the basal and apical surfaces of the Txnip.CS.LLAA mutant overexpressor but it seems to be only on the apical surface. Please mention and discuss this.

The “bright band” of GLUT1 is actually on the basal surface, not apical. We have updated the figure and legend to indicate this. With Best1-Txnip addition, the apical surface of the RPE collapses onto the photoreceptors, which also have GLUT1. These structural changes make it hard to tell how much GLUT1 is present on the apical surface of the RPE, when they express the CS.LLAA mutant compared to the WT Txnip.

Reviewer #2 (Recommendations for the authors):The manuscript includes extensive data. However, several points require clarification and the text and figures need consistency. Specific comments/suggestions are as follows:The authors concluded previously that cone starvation and degeneration are due to shortage of glucose in four animal models of RP (Punzo et al., 2009). However, except rho-/-, which is not the focus in this manuscript, the rest three animal models were not used in this study. It is unclear whether rd1 and rd10 also have shortage of ATP and how severe the shortage is as compared to the WT, or it is simply more ATP can better maintain cone survival.

We did not use Pde6γ^-/-^ (as we did in Punzo et al. 2009), because its rod-cone degeneration rate is similar to that of Pde6β^-/-^, i.e. the *rd1* strain. We also did not use the transgenic P23H strain (as in Punzo et al.) or a more recent P23H knock-in strain from the Palczewski Lab, due to the extremely slow rod-cone degeneration rate. We used the *rd10* strain, which carries a Pde6β hypomorphic mutation instead of a null mutation. The rod-cone degeneration speed of *rd10* is intermediate between those of *rd1* and *Rho*^-/-^ (Figure 1—figure supplement 1A, originally Extended Data Figure 1a), thus allowing us to investigate mutant strains with 3 rates of degeneration. Given that all of these mutant strains have a rod-specific mutation, and cone degeneration is secondary, our hypothesis throughout all of this work is that we are dealing with a non-autonomous set of problems for genetically wildtype cones. In Punzo et al. 2009, as well as several other gene therapy publications from our lab (Xiong et al. 2015, Wang et al. 2019, 2020, Wu et al. 2021, Chinchore et al. 2019), any treatment that benefits *rd1* also benefits the *rd10* and *Rho*^-/-^ strains. Also, as we showed in Punzo et al., the changes in cones across the original 4 strains that we used had the same hallmarks, which suggests that the mechanisms leading to cone death are common across the strains. The additional time and expense of carrying out the Txnip experiments on P23H did not seem worthwhile in light of these findings.

The authors claim the treatment "correlates with the presence of a healthier mitochondria". As shown in Figure 5a, although the size of mitochondria was bigger in the treated groups compared to the control, those mitochondria did not seem to harbor a typical morphology. More transmission electron microscopy (TEM) images of different magnitudes are needed to show the localization of these mitochondria as well as the morphology of more mitochondria. Without the images from WT, it is hard to evaluate whether it is due to strain of the mice or other factors, or how a healthy mitochondrion should look like. Likewise, a much higher membrane potential in mitochondria does not necessarily mean healthier mitochondria. Additionally, the authors should show the mitochondria morphology of P50 rd1 as well, as that's the end point of the whole experiments and can provide useful information on the duration of the treatment.

Please see our response to the Editor (point #2) above. We have supplied more TEM images to address the question of mitochondrial morphology (Figure 5—figure supplement 2B-D). We are not claiming that Txnip-treated RP cones, or their mitochondria, look like wild type, only that they show several features of mitochondrial morphology and function that correlate with greater health. The TEM data show that they are larger in cross-section, the mitoRFP shows that they have a greater membrane potential and appear more “numerous”, the JC-1 dye aggregation shows greater membrane potential. All of these assays are standard assays used to investigate mitochondrial status. Furthermore, PercevalHR imaging showed that there is more ATP generated by Txnip treated cones, which is most likely due to better mitochondrial function, and that this is true when lactate is supplied, which is dependent upon Ldhb. In keeping with these data, Txnip C247S, provided for more cone survival, as well as had correlated benefits to cone mitochondria. Our RP cone RNA-seq data showed more ETC mRNAs, in correlation with the other assays of mitochondria. We did not rely only on TEM, but performed several independent assays to investigate the effects of Txnip on mitochondria.

Although the flat mount images in this manuscript show the overall cell survival in the whole retina, immunostaining of retinal sections is needed to address important questions related to retinal functions such as the structure of the retina, the morphology of the survival cone photoreceptors and the subtypes of the surviving cones, i.e., whether Txnip can maintain both S- and M- cones. The authors mentioned "this combination also preserved the RP cone outer segments" (line 312); yet, it is hard to evaluate the outer segment by flat mount, although a slight difference may be noticeable between the two groups.

We recently harvested and sectioned some P122 rd10 eyes transduced with RedOTxnip or RedO-Txnip + Best1-Nrf2. These sections are stained for S- or M- opsin separately. It showed that Txnip preserved expression of both S- and M- opsins, and Txnip + Best1-Nrf2 led to better cone outer segment structures with enriched S- and M- opsin (new Figure 8—figure supplement 1A). Again, we do not claim wildtype cone outer segment morphology, only that the combination of RedO-Txnip + Best1-Nrf2 gave the best outer segment morphology of degenerating cones that we have seen.

The automated method for cone quantification (1/2 radius cell count) that counts H2BGFP-positive cones in the central half of the retina (previous calculation of the diameter of flat mount retinas) raises some concerns. Is the area of the retina in the 1/2 radius the same between samples? Not all flat-mounted retinas look the same (differences due to retina dissection), and this can influence the area used to count cones.

When we began studies of RP cone rescue, and needed to quantify cones, we used multiple methods of IHC for cone markers (PNA, anti-opsin, anti-cone arrestin, cone lacZ transgene, cone RNA quantification). Due to lack of good morphology and a reduction in expression of cone genes in sick cones, we found all of these methods to be difficult and the IHC methods to be subjective. However, the H2BGFP protein seems to be quite stable, as predicted if it is localized within chromatin which is not turning over at an appreciable rate in cones. The fact that it shows bright nuclear staining made quantification, by eye or by an automated program, to be the most robust method that we have used. The specificity of expression by the RedO promoter, after rods are gone (some of which can express at a low level), made it possible to count from the flat mount. We bench marked this method to FACS quantification using anti-cone arrestin (see Wang et al. 2020). In addition, we provide data here showing the correlation with FACS quantification (new Figure 1—figure supplement 2C).

However, to address the point of variability in dissections and flat mounting, each retina is outlined and then the approximately central point of the optic nerve, which is unambiguous, is used to draw the central 50%. We have not shown the best images in our figures, but have shown typical images. The dissections and flat mounting procedure are quite reproducible as we have become quite proficient with these methods. There is certainly some natural variation of area among the samples, which is why we use the ½ radius as a normalization of each retina’s area. In addition, please note that the number of retinas for each experiment is quite high, and we include controls from the same animal (e.g. one eye injected with control and the other eye with Txnip). The P50 FVB *rd1* control cone count is usually consistently around 4,000 in the central region using this method.

In addition, is this method counting all the cones present in the sample? Are all cones H2BGFP-positive? What is the viral transduction efficiency (alone and co-injecting two AAVs)? Is it the same between samples? Are authors using z-stack confocal images for quantification?

The injection of 2.5 x 10^8^ vg/eye RedO-H2BGFP labels 20% of cones by our own estimation using WT mouse cone counts from Jeon, Strettoi and Masland (1998, J Neurosci). We have added this information in the Results section, where we first describe this labeling.

Because there is no difference between control and Txnip in P20 *rd1* cone counting (Figure 1B, and new Figure 2—Figure supplement 1C), we do not think there is any difference in H2BGFP transduction rate alone vs. co-injection. We do not believe that we need to label every cone, but to reproducibly sample the existing cone population. It is also important to note that we may miss rescued cones with this method, when a cone is infected with Txnip but not H2BGFP. This is a loss that we are willing to take, as it deflates, rather than inflates, the rescue. The trade-off in the ease and confidence in scoring the H2BGFP nuclei is one we are quite willing to make.

As described in Methods-histology, we used a commercial fluorescence microscope with a 10x objective to capture and stich the images. By focusing on the residual ONL, which is only one cell layer thick, we do not need z-stacks to capture all the H2BGFP labeled cones in the RP retina. Overall, we believe the RedOpsin-H2BGFP counting method could become a useful method to assay RP cone survival for the field. We have deposited our MATLAB code for cone quantification on Github (https://github.com/sawyerxue/RP-cone-count), and it will be released to the public soon.

It is appropriate to quantify density of cones (number of cones/mm2) in defined areas of interest, which should be the same for all the samples. To achieve this, a proper orientation of the whole-mounted retina is necessary, and this information is lacking in the images included in the manuscript.

By using a computer program to count all of the labeled cones within a ½ radius circle of the entire retina, we were able to screen many vectors and many retinas, which alleviates concerns of measuring cone rescue only in a specific, perhaps selected or biased, area. In another study in our lab, we are quantifying the survival of peripheral cones, where we do see a bias in survival of the dorsal cones. In the current study, by counting all of the cones in the central retina and using a fairly large sample size for all studies, we do not see the need to orient the retina. The reproducible number of cones in the central retina in the control group (approximately 4,000) again suggests that random orientations do not create biases or irreproducible counts.

The authors could consider the quantification of "holes" or "craters" in the cone mosaic as an indicator of differential treatment effect.

As described in the text, craters are far more common in the FVB strain of the *rd1* mutant. We assayed craters in 4 other inbred strains of mice that carry the *rd1* allele and saw very few craters (Punzo and Cepko, unpublished). The cone counts do not correlate with the amount of these craters. Interestingly, we found that treatment of the RPE with AAV-Best1Nrf2 almost completely eliminated the FVB craters (Wu et al. 2021), without much benefit to cones. This suggests that the craters originate with problems in the RPE (oxidative damage, inflammation, perhaps metabolic problems) as Nrf2 may affect any or all of these problems. The lack of a benefit to cones is interesting, as we would have predicted an effect.

The authors do not quantify the level of Txnip expression in transduced retinas. It would be important to know the level of overexpression they have achieved and compare it with the expression in untreated RP retinas. In addition, it would be relevant to know if treatment effects correlate with Txnip expression levels.

Please see our response to the Editors (point #1) above on the Txnip antibody issues for IHC. The RNA level of Txnip treated vs. control cones can be found in a new Figure 5— source data 3. Due to the fact that cones normally express almost no Txnip mRNA, the increase of Txnip mRNA level is huge when adding RedO-Txnip.

OMR (based on head movements of mice observed by a human observer) can't be directly equated to visual acuity (see PLoS ONE 8(11): e78058, 2013; J Neurophysiol 118: 300-316, 2017). One needs to interpret such data with caution. Is it possible to do multi-focal ERG?

We agree with the reviewer that the OMR is sometimes not equal to visual acuity, because OMR can be used to measure different visual performances. However, visual acuity is one of them, and our use of it here did measure this parameter. A common visual performance that can be measured by OMR is contrast sensitivity, which we did not test here. By definition, visual acuity is equal to “spatial frequency threshold”, which is exactly the visual performance that we measured using OMR in this and previous studies (Xiong W et al., 2015; Wang SK et al., 2019, 2020). Based on these studies, we believe that the visual acuity measured by OMR reflects areas of improved cone function, which may reflect increased cone density and health in e.g. central retina.

We are not sure about the relevance of the two references provided by the Reviewer here, because both references support OMR as a reliable method of quantifying visual performance, including acuity. In PLoS ONE 8(11): e78058, 2013:

“Exemplary, we show that automatically measured visual response curves of mice match the results obtained by a human observer very well. The spatial acuity thresholds yielded by the automatic analysis are also consistent with the human observer approach and with published results. Hence, OMR-arena provides an affordable, convenient and objective way to measure mouse visual performance.”

J Neurophysiol 118: 300-316, 2017:

“We provide the first direct comparison of OMR and OKR gains (head or eye velocity/stimulus velocity) and find that the two reflexes have comparable dependencies on stimulus luminance, contrast, spatial frequency, and velocity.”

Multi-focal ERG is for animals with large eyes, including humans. For mice, which have tiny eyes, the Reviewer might be referring to focal ERG instead. Since we have the entire retina infected with P0 subretinal injection, i.e. not a partial infection as in human subretinal injections, we do not see the advantage of using focal ERG vs. full-field ERG. We observed no difference between control vs. Txnip in full-field ERG in the P40 rd10 mice (see new Figure 1—figure supplement 2E). Because ERG reflects overall phototransduction across the entire retina (Xue Y… Kefalov VJ, 2015a,b, 2017, 2020), but not improved functional cone density, i.e. improved visual acuity where cone survival is improved in the center, this result of the ERG does not argue against the positive OMR visual acuity results. Because cone loss is greatest in the central retina at the ages measured, the OMR result likely supplies the most robust measure of an improvement. In addition, this ERG result at least suggests that Txnip does not make the RP cone phototransduction worse than the control, which is an important consideration for therapeutic applications.

The authors confirmed the effect of Txnip on cone survival using three different animal models. However, in some follow-up experiments, the authors used these models interchangeably and it leads to confusion. For example, while rd1 is the main focus of the manuscript, the authors did not show the rescued visual acuity of these mice. Even if there is no effect of Txnip on visual acuity in these mice, the authors should still mention this to provide some information of the effectiveness of the treatment on some fast-degenerating mouse models. In addition, the authors used rd10 and rho-/- as well in the manuscript. Did the authors test the mechanisms they found using rd1 on rd10 and rho-/-? These data would be helpful to discriminate whether the switch of energy source by Txnip in the survival of cone cells is specific to rd1.

We added a sentence in Methods: “Optomotor tests were conducted on *rd10* and *Rho*^-/-^ mice, but not with *rd1* strain, which loses vision at a very early age before any meaningful test can be performed.”

We have done RNA-seq with P90 *Rho*^-/-^ mice treated with Txnip vs. control (Figure 5— figure supplement 1A, originally Extended Data Figure 5a), and found a shared upregulation of mitochondrial ETC gene upregulation with *rd1* cones (Figure 5—source data 1, originally Supplementary Table 2). As explained above to Reviewer #2 (first point), cone degeneration shares a similar pattern across strains (Punzo et al., 2009). Given the improvement in *rd10* and *Rho*^-/-^ vision via the OMR assay, the RNA-seq results, and the common hallmarks of degeneration across all of the strains, we did not believe it necessary to perform more assays on the *rd10* and *Rho*^-/-^ strains. *Rd1* was the preferred strain as it is more rapid than the other strains.

Figure 8a and 8b described the effect of vector combinations on cone survival in rd1, whereas Figure 8c actually mentioned the effect on outer segment in rd10. The authors did not even mention this switch of animal models in the text. The effect on outer segment in rd10 mice does not mean it would surely give the same effect in rd1 mice. It is unclear whether there was no improvement of outer segment in treated rd1 mice, or the vector combination did not have any effect on cone survival in rd10 mice.

The results from the combination treatment are quite new, and we were excited to include them here. However, they are not central to the overall finding that Txnip rescues cones in the 3 strains where we assayed cone survival (Figure 1), nor any of the other data on the mitochondria or Ldhb, or Parp1 KO. They are a result from our ongoing effort to combine treatments, as described above. But to address the question raised by the Reviewer, we have not examined the *rd1* strain with anti-opsin with this combo, nor did we measure cone survival in P130 *rd10* mice.

Reviewer #3 (Recommendations for the authors):In order to strengthen this manuscript, the following suggestions should be addressed:1. Did the authors observe the difference in mice with or without NNT deletion? Is there an NNT deletion in rd1, rd10, parp1-/-, and rho-/- mice?

We thank the reviewer for bringing this to our attention. The *Nnt* null mutation (MGI: 3626282) was observed only in the C57BL/6J background, and only *rd10* mice in this study are on C57BL/6J background. We confirmed the presence of this mutation in our C57BL/6J and rd10 colonies by genotyping through Transnetyx, and found no *nnt* mutation in other strains (i.e. rd1, *Rho*^-/-^, *Parp1*^-/-^, CD1 or BALB/c). This information has been added in the Animals section of Methods.

Since *rd10* mice are also responsive to Txnip rescue, NNT presence or absence does not affect the ability of Txnip to rescue. This result provides additional support that Txnip is a promising gene-agonistic therapy candidate for RP patients with various genetic backgrounds.

2. Extended Data Figure 1b and 1d: to show AAV8-RO1.7-GFP expression in cone photoreceptors, the Arr3 antibody will be better than PNA since PNA could only express in the cone outer and inner segments. Extended Data Figure 1b showed some GFP in the cytoplasm, is this the H2B-EGFP?

To reveal the full cone structure in different conditions, a new Figure 1—figure supplement 2A has been added with ARR3 staining. However, we believe PNA staining, which labels cone extracellular matrix including cone pedicles, is a more sensitive readout for toxicity to cones than ARR3 staining (Xiong, Wu, Xue et al., 2019, PNAS).

The GFP signal in Extended Data Figure 1b (now Figure 1—figure supplement 1B) is from the GFP-Txnip fusion protein, which does not have the H2B tag. We have updated all figures with in-panel color-code info to avoid any confusion to other readers.

3. Figure 1b: are these numbers within the 1/2 radius or within selected small boxes? If these numbers are from selected small boxes, how many small boxes from each were selected? The authors should explain how they selected the small boxes as there is some variation on the dorsal retina or ventral retina. For example, in P130 rd10, the cone cell densities are quite different in the four leaves of the inner circle. Same question for the P50 rd1, there is some area with a patchy absence of cone cells. If the numbers are within 1/2 radius, why use 1/2 radius rather than 2/3 or 1/3 radius?

The small boxes in Figure 1a were enlarged in Extended Data Figure 1c (now Figure 1—figure supplement 1C) to demonstrate the MATLAB program’s results of cone counting. Rather than just count the cones in the small boxes, Our MATLAB program counted all of the cones within the ½ radius of the entire retina (as a circle) to avoid any uneven degeneration in the four leaves.

To avoid any confusion, we have updated the Figure legend of Figure 1a to better explain the small boxes, which now reads: “All H2BGFP-labeled cones were counted within the central retina defined by the ½ radius (i.e. not just the cells from the small boxes).”

4. It would be helpful if the authors explained the reason behind each time point, i.e., P50 for rd1, P130 for rd10, and P150 for Rho -/-.

We have been using these ages as a standard for cone degeneration based on our previous assays of cone degeneration across time in the various strains (Punzo el al., 2009; Xiong et al., 2015; Wang SK et al. 2019, 2020). We have added an explanatory sentence in the text, and the progression over time is illustrated in Figure 1—figure supplement 1A.

5. Figure 5i: the number of cone cells at 1/2 radius in the whole mount retina looks quite similar, but it seems the number of cone cells in the periphery of control retina is more than the number in the +Ldhb retina.

As we have mentioned in Methods – Automated cone counting, the brightness in certain parts of the retina does not correlate with the number of cells. We believe our MATLAB quantification method produces the most reliable readout for the number of surviving cones. The central ½ radius is the best place to count, as the peripheral retina can have a portion of cones that live an extraordinary period of time, as we are attempting to understand. Moreover, the central retina is more critical for visual acuity than the peripheral retina.

6. Bracket "[]" in Figure 4, and curly bracket "{}" in Figure 5 indicate sample size. Are these the sample sizes of the image/retina or samples sizes of the mice?

This information is in the corresponding figure legends. Here briefly, in Figure 4b: “[]” is the number of images taken from regions of interest of multiple retinas. In Figure 5b: The number in the curly bracket “{ }” indicates the sample size, i.e. the number of mitochondria from multiple cones of >one retina for each condition. We duplicated this information to all other figures where it applies.

7. Figure 5: it is unclear how reliable the Mt size is, because the EM images are not SBF-SEM. The shape and morphology of mitochondria is in 3D and the EM images are taken from a different orientation and thus may have different results.

We agree with the reviewer that SBF-SEM is a better technology for mitochondria morphology. However, when we obtained the TEM imaging from multiple samples, it was very striking that the larger mitochondrial cross sections were in WT Txnip and Txnip.C247S treated *rd1* cones rather than in controls, as reflected by the distribution of outliers in these group (Figure 5B). We have added a new Figure 5—figure supplement 2 showing more mitochondrial images from different conditions.

8. Did the authors observe any difference in the number of mitochondria in Txnip-treated retina?

As shown in the new Figure 5—figure supplement 2, we did not observe any obvious difference in mitochondria numbers (also see response to Reviewer #2). As the Reviewer pointed out, TEM might not be the best technology to answer this question. From our flatmount mitoRFP images (Figure 5—figure supplement 1C,D, originally Extended Data Figure 5c,d), there is an increase in mitoRFP density in the central retina by Txnip. This likely reflects an increase of mitochondrial density in this region.

9. Does Txnip increase mitochondrial biogenesis?

As explained in section# 3 to the Editors, please refer to the new Figure 5—figure supplement 2A and new Figure 5—source data 2 for RNA-seq data on Txnip treated WT cones.

10. It would be better if the authors could clarify the meaning of n.d. in "Wu et al., n.d.".

This work was previously in press, and recently published. We have updated this reference.

11. Line 336-337: it remains unclear what the role of anti-oxidative stress is in the Txnip-mediated cone rescue effect. It would be helpful if the authors added an experiment to demonstrate the changes in ROS in Txnip (wt and C237S)-mediated activities.

The Txnip WT allele is believed to decrease the anti-oxidative stress function of thioredoxin through binding that requires the Txnip C247 residue , i.e. the WT allele should *increase* oxidative stress. See the discussion above regarding the role of Txnip in oxidative stress. It is clear that more work needs to be done to understand our findings relative to oxidative stress.

12. Line 353: it would be helpful if the authors could explain why the Txnip rescue did not depend on ketolysis or ß-oxidation.

While we cannot know the answer to this question, we did speculate, in the original Discussion: “An important factor in the reliance on Ldhb could be the availability of lactate, which is highly available from serum (Hui et al., 2017). Lactate could be transported via the RPE and/or Müller glia, and/or the internal retinal vasculature which comes in closer proximity to cones after rod death. Ketones are usually only available during fasting, and lipids are hydrophobic molecules which are slow to be transported across the plasma membranes. Moreover, lipids are required to rebuild the membrane-rich outer segments, and thus might be somewhat limited.”

13. Line 487-488: "The GFP fluorescence intensity served as a fast 488 and direct read out of the knockdown efficiency of these shRNAs." The GFP after IRES can sometimes too weak to produce a fast and direct readout. Do the authors have the image/a figure on this?

Yes, these images were supplied in the original version, in Extended data Figure 8-11 (now Figure 2—figure supplement 2, Figure 3—figure supplement 2-4).

14. It is unclear whether the control group in this study refers to the non-injected eye or to saline injections

In Methods- subretinal injection section, we have added a sentence that reads: “All of the control groups in this study refer to AAV reporter (e.g. H2BGFP or PercevalHR) injection alone.”